# Machine learning-assisted elucidation of CD81–CD44 interactions in promoting cancer stemness and extracellular vesicle integrity

Erika K Ramos[1,2], Chia-Feng Tsai[3], Yuzhi Jia[1], Yue Cao[4], Megan Manu[1], Rokana Taftaf[1,2], Andrew D Hoffmann[1], Lamiaa El-Shennawy[1], Marina A Gritsenko[3], Valery Adorno-Cruz[1], Emma J Schuster[1,2], David Scholten[1,2], Dhwani Patel[1], Xia Liu[1,5], Priyam Patel[6], Brian Wray[6], Youbin Zhang[7], Shanshan Zhang[8], Ronald J Moore[3], Jeremy V Mathews[8], Matthew J Schipma[6], Tao Liu[3], Valerie L Tokars[1], Massimo Cristofanilli[7,9], Tujin Shi[3†], Yang Shen[4†], Nurmaa K Dashzeveg[1†], Huiping Liu[1,7,9\*†]

[1]Department of Pharmacology, Northwestern University, Chicago, United States; [2]Driskill Graduate Program in Life Science, Feinberg School of Medicine, Northwestern University, Chicago, United States; [3]Biological Sciences Division, Pacific Northwest National Laboratory, Washington, United States; [4]Department of Electrical and Computer Engineering, TEES-AgriLife Center for Bioinformatics and Genomic Systems Engineering, Texas A&M University, College Station, United States; [5]Department of Toxicology and Cancer Biology, University of Kentucky, Lexington, United States; [6]Quantitative Data Science Core, Center for Genetic Medicine, Northwestern University Feinberg School of Medicine, Chicago, United States; [7]Department of Medicine, Hematology/Oncology Division, Feinberg School of Medicine, Northwestern University, Chicago, United States; [8]Pathology Core Facility, Northwestern University, Chicago, United States; [9]Robert H. Lurie Comprehensive Cancer Center, Feinberg School of Medicine, Northwestern University, Chicago, United States

\*For correspondence:
huiping.liu@northwestern.edu

†These authors contributed equally to this work

**Abstract** Tumor-initiating cells with reprogramming plasticity or stem-progenitor cell properties (stemness) are thought to be essential for cancer development and metastatic regeneration in many cancers; however, elucidation of the underlying molecular network and pathways remains demanding. Combining machine learning and experimental investigation, here we report CD81, a tetraspanin transmembrane protein known to be enriched in extracellular vesicles (EVs), as a newly identified driver of breast cancer stemness and metastasis. Using protein structure modeling and interface prediction-guided mutagenesis, we demonstrate that membrane CD81 interacts with CD44 through their extracellular regions in promoting tumor cell cluster formation and lung metastasis of triple negative breast cancer (TNBC) in human and mouse models. In-depth global and phosphoproteomic analyses of tumor cells deficient with CD81 or CD44 unveils endocytosis-related pathway alterations, leading to further identification of a quality-keeping role of CD44 and CD81 in EV secretion as well as in EV-associated stemness-promoting function. CD81 is coexpressed along with CD44 in human circulating tumor cells (CTCs) and enriched in clustered CTCs that promote cancer stemness and metastasis, supporting the clinical significance of CD81 in association with patient outcomes. Our study highlights machine learning as a powerful tool in facilitating the molecular understanding of new molecular targets in regulating stemness and metastasis of TNBC.

## Editor's evaluation

This research innovatively depicted that CD81 could form interaction with CD44 and contributes to the self-renewal, and metastatic capacity of breast cancer stem cells. Through large amounts of validation, either through computational or experimental methods, the conclusions are firmly held back that CD81-CD44 interaction is indispensable for circulating tumor cell dissemination and metastasis. The findings have broadened the knowledge of the molecular regulatory network of breast cancer cell stemness and may be significant for scholars who are experts in cancer biology.

## Introduction

Breast cancer killed nearly 0.7 million people worldwide in 2020 with metastasis accounting for 90% of deaths (*Sung et al., 2021*). Negative for estrogen receptor, progesterone receptor, and HER2 amplification, triple negative breast cancer (TNBC) comprises 10–15% of newly diagnosed breast cancer cases and is highly metastatic with low long-term survival (*Perou et al., 2000*; *Sørlie et al., 2001*; *Rakha et al., 2007*; *Haffty et al., 2006*). TNBC preferentially metastasizes to the visceral organs such as the lungs, liver, and brain (*Foulkes et al., 2010*; *Dent et al., 2007*). Growing evidence suggests tumor-initiating cells, or cancer stem cells with self-renewal and regenerative reprogramming capacity, are the underlying cause of cancer relapse, therapy resistance, and distant dissemination (*Diehn et al., 2009*; *Li et al., 2008*; *Al-Hajj et al., 2003*; *Charafe-Jauffret et al., 2010*; *Giordano et al., 2013*; *Idowu et al., 2012*; *Lin et al., 2012*; *Owens and Naylor, 2013*; *Liu et al., 2010*; *Liu and Wicha, 2010*; *Korkaya and Wicha, 2009*; *Adorno-Cruz et al., 2015*; *Ramos et al., 2017*) with measurable clinical impact on patient outcomes (*Liu et al., 2010*; *Bhola et al., 2013*; *Calcagno et al., 2010*; *Phillips et al., 2006*; *Creighton et al., 2009*; *Shats et al., 2011*; *Lathia et al., 2020*; *Charafe-Jauffret et al., 2010*; *Li et al., 2008*; *Idowu et al., 2012*; *Lin et al., 2012*; *Owens and Naylor, 2013*). However, the cellular and molecular mechanisms contributing to tumor cell spreading and regenerative metastasis are yet to be fully understood. Among the biomolecular mechanisms, protein functions, and their networks are the backbones directly related to cellular phenotype and performance. Machine learning and deep learning have transformed protein structure modeling and greatly facilitated the understanding of molecular networks and therapeutic development (*Karimi et al., 2020*; *AlQuraishi, 2021*; *Baek and Baker, 2022*).

Circulating tumor cells (CTCs) with genetic and/or epigenetic alterations (*Fernandez et al., 2014*; *Gkountela et al., 2019*) are tumor cells that shed from cancer lesions and circulate within the blood or lymphatic vasculature before seeding with a very low frequency to distant organs for metastatic tumor regeneration. Detection of CTC events on the CellSearch platform is associated with patient outcomes in which multicellular CTC clusters predict worse prognosis and mediate metastasis in a 20- to 100-fold higher efficiency than single CTCs (*Mu et al., 2015*; *Aceto et al., 2014*; *Liu et al., 2019*). Our recent work demonstrated that the breast tumor-initiating cell marker CD44 is highly enriched in CTC clusters and predicts an unfavorable overall survival of patients with breast cancer, especially TNBC (*Liu et al., 2019*). CD44 mediates homophilic interactions for CTC cluster formation as a mechanism to promote self-renewal (mammosphere formation) and polyclonal metastasis in TNBC (*Liu et al., 2019*). While most of the previous studies focus on the genome, epigenome, or transcriptome alterations in clustering tumor cells (*Gkountela et al., 2019*; *Aceto et al., 2014*; *Ting et al., 2014*), little is directly linked to proteome alterations. We compared mass spectrometry proteomic profiling of clustering and non-clustering TNBC patient-derived xenografts (PDXs) tumor cells and identified CD81, a tetraspanin protein enriched in extracellular vesicles (EVs) (*Kowal et al., 2016*; *Mathieu et al., 2021*), as one of the altered proteins upon CD44 depletion.

One of the objectives of the study is to elucidate the molecular mechanisms underlying the regulatory partnership between CD44 and CD81 in self-renewal, CTC clustering, and progression of TNBC. However, the protein structure of CD44 has not been experimentally solved. To facilitate unveiling the molecular basis of CD44 and CD81 protein networks and testing the hypothesis that CD44 and CD81 interact with directly, we employed machine learning-based protein structural modeling approaches such as iTasser (*Yang et al., 2015*), ClusPro (*Kozakov et al., 2017*), and Bayesian Active Learning (BAL) (*Cao and Shen, 2020*). To discover CD44- and CD81-regualted downstream pathways, we also performed mass spectrometry-based phosphoproteomic profiling and machine learning-assisted bioinformatic analyses using protein identification platforms (*Kong et al., 2017*; *Teo et al., 2021*) and

pathway analyses such as DAVID (*Jiao et al., 2012*), STRING (*Szklarczyk et al., 2021*), and Cytoscape (*Shannon et al., 2003*), which revealed endocytosis as one the top pathways regulated by not only CD81, but also CD44, which is unexpected.

Endocytosis is closely related to biogenesis of EV such as exosomes. Most of cell-secreted EVs (30–1000 nm), if not all, are characterized by presence of membrane structure, enriched protein markers such as CD81, CD63, CD9, and TSG101, and other components like lipids, nucleic acids, and metabolites (*Kowal et al., 2016*; *Mathieu et al., 2021*; *Bobrie et al., 2011*; *Théry et al., 2006*). The biogenesis pathway of large EVs (such as microvesicles) via plasma membrane shedding is distinct from that of small EVs (such as exosomes). The latter is generated from the invagination of the plasma membrane to form endosomes that continue to invaginate and mature into multivesicular bodies destined for either lysosome degradation or fusion with the plasma membrane and release as exosomes (*Bobrie et al., 2011*). EVs have emerged as key players in intercellular communications in both physiological and pathological settings, including cancer development (*Iero et al., 2008*; *Kahlert and Kalluri, 2013*) and distant organ-specific metastasis (*Hoshino et al., 2015*). However, the role of EVs in cancer development and its association with CD44 and CD81 functions remain unclear. This work aims to determine the role of CD81 in EVs, tumor initiation, and metastasis. Following the widely adopted nomenclature in the EV field (*Théry et al., 2018*) and considering the technical limitations in exosome isolation that contains heterogenous EV populations, we therefore utilize 'EVs' or small EVs to describe the enriched exosomes in our study.

Here, using the TNBC PDXs established and maintained in mice (*Liu et al., 2010*; *Liu et al., 2019*), human cell lines, and mouse tumor models with genetic modulations such as knockdown (KD) and knockout (KO), we examined the importance of CD81 in TNBC progression with a novel mechanistic link to CD44 interactions and self-renewal which is enhanced by uptake of cancer exosomes and coupled with endocytosis and metabolic pathways, and identified previously unknown tumor-intrinsic functions of CD81 in promoting tumor clustering and lung metastasis.

## Results

### CD81 promotes mammosphere formation and interacts with CD44

We previously found that TNBC PDXs express splicing variants of CD44 (CD44v), whereas MDA-MB-231 cells predominantly express standard CD44 (CD44s), both contributing to CTC cluster formation and cancer metastasis (*Liu et al., 2019*). To better understand the proteomic alterations and mechanistic regulation of self-renewal and tumor cluster formation, we conducted global mass spectrometry analyses of TNBC cells, including clustering and non-clustering PDX tumor cells (CD44+ and CD44− cells) as well as MDA-MB-231 cells of wild-type (WT) and *CD44* KO, which pooled populations were generated via multiple guide RNAs and CRISPR-Cas9 technique (*Liu et al., 2019*). Based on two paired comparisons with PDX cells freshly isolated from mice, including flow-sorted CD44+ versus CD44− cells (mimicking circulation) and si*CD44*-mediated KD versus siRNA control cells that clustered ex vivo (*Liu et al., 2019*), we identified 38 overlapping proteins differentially expressed in these cells with a more than twofold change (*Figure 1—figure supplement 1A*, *Supplementary file 1*). In addition to PAK2 as a previously reported CD44 target (*Liu et al., 2019*), one of the top listed proteins was CD81, an EV marker and tetraspanin family membrane protein, reduced in sorted CD44− or CD44 KD cells. Consistently, CD81 was present in the TNBC PDX (CD44v) cells but decreased in CD44KO PDXs (*Figure 1—figure supplement 1B*). However, in MDA-MB-231 cells that express CD44s, CD81 was only temporarily downregulated in CD44KO cells in suspension but comparable between adherent CD44 WT and KO cells, as detected by immunoblotting (*Figure 1—figure supplement 1C*), suggesting context-dependent alterations of CD81 by CD44 in TNBC cells.

Since CD44 is known to promote tumor initiation and metastasis in breast cancer (*Liu et al., 2019*), we first tested the role of CD81 in self-renewal in TNBC by assessing its impact on mammosphere formation, cell growth, and pluripotency-related genes and proteins after gene KO, which was generated as pooled populations with two *CD81* gRNAs using the CRISPR-Cas9 approach (*Figure 1—figure supplement 1C*). In addition to CD44KO and CD81KO, a combined CD44/CD81 double KO (dKO) cell line was also made for phenotypic analyses (*Figure 1—figure supplement 1B*). In comparison to the WT MDA-MB-231 cells, pooled populations of CD44KO, CD81KO, and dKO cells show similar, diminished capability for mammosphere formation, with fewer and smaller spheres where the

dKO did not show much additional effects than single KOs (*Figure 1A, B*), suggesting both CD81 and CD44 are required for optimal self-renewal with similar functional importance.

Using multiple mouse *Cd81* gRNAs and CRISPR-Cas9 approach, we also created pooled Cd81KO populations in mouse 4T1 TNBC cells which displayed fewer mammospheres compared to the WT cells, after being seeded with a low number of cells in stem cell medium in suspension, (*Figure 1—figure supplement 1D–F*). To assess human CD81 and mouse Cd81 functions in cell growth, we employed IncuCyte time-relapse imaging to monitor the cell confluence in adherent culture. Both CD81KO and Cd81KO cells showed a slightly less confluency compared to respective WT cells (*Figure 1—figure supplement 2A,B*), indicating that effects of CD81KO on diminished self-renewal (mammosphere formation) are beyond a slightly altered cell growth or proliferation. We then measured the expression levels of stem-cell signature genes and/or proteins in these cells. Like CD44KO cells, CD81 KO populations decreased protein levels of breast tumor-initiating markers including OCT4, NOTCH1, and phosphoSTAT3 (*Figure 1—figure supplement 2C*).

To determine whether the two membrane proteins CD44 and CD81 influence each other's cellular localization or distribution, we further performed immunofluorescence staining of human TNBC cells with CD44KO or CD81KO, adherent and in suspension (for clustering activity analysis). CD44 was observed mostly on the cytoplasmic membrane in WT MDA-MB-231 cells, both adherent and in suspension, but the intracellular CD44 accumulated specifically in the cells in suspension (p = 0.03) (*Figure 1C,D*). In the WT group, ~50% of adherent cells and 70% of suspension cells showed an average of 14–16% of partial colocalization between CD44 and CD81 on the cytoplasmic membrane, with CD81 mainly presented at the interface of the clustered tumor cells in suspension (*Figure 1C,D*, *Figure 1—figure supplement 2D*). The KO of CD44 or CD81 altered the localization of CD81 or CD44, respectively, with disrupted localization to the membrane in suspension but increased staining at intracellular loci of adherent cells (*Figure 1C,D*, white arrows in top panels). Meanwhile, CD44KO or CD81KO further weakened the detection of both in either intracellular loci or surface membrane in suspension cells (*Figure 1D*, white arrows in bottom panels). These data demonstrate that cell detachment influences the cellular localization of CD81 and CD44 which facilitate each other's membrane presentation and subsequent co-localization, especially at the interface of neighboring and clustering tumor cells.

## Machine learning-assisted dissection of CD81–CD44 interactions via extracellular regions

To further examine if CD81 directly interacts with CD44 and to identify possible molecular regions responsible for proposed CD44–CD81 interactions in TNBC cells, we employed machine learning-assisted protein structure modeling (*Cao and Shen, 2020*) and co-immunoprecipitation (co-IP) tests. Based on previous structural studies on CD44 (*Kawaguchi et al., 2020*) and CD81 (*Zimmerman et al., 2016*) as well as computational programs iTasser (*Yang et al., 2015*), ClusPro (*Kozakov et al., 2017*), and BAL (*Cao and Shen, 2020*), we first analyzed CD44 and CD81 protein sequences and possible dynamics of protein–protein interaction models. As shown in *Figure 1E*, the hotspot residues in warm colors (red and yellow) predicted to be involved in the interactions are located in domains I and II of CD44, which also contribute to CD44–CD44 homophilic interactions (*Liu et al., 2019*; *Kawaguchi et al., 2020*), and the extracellular loop of CD81, which links its third and fourth transmembrane domains, with an estimated free energy of binding at −12.23 kcal/mol. As expected, after being immunoprecipitated (IP) by anti-CD44 antibody, the endogenous CD81 and CD44 proteins were simultaneously observed within a membrane–fraction protein complex from the lysates of TN1 PDXs (expressing CD44v) and MDA-MB-231 cells (expressing CD44s), but not CD44KO cells (*Figure 1F*). We then overexpressed tagged CD44-Flag and CD81-HA in CD44⁻ HEK293T cells, in which CD44 was shown in multiple forms at distinct molecular weights (*Figure 1—figure supplement 2E*), possibly due to variable glycosylation patterns as we previously reported (*Kawaguchi et al., 2020*). Both CD44-Flag and CD81-HA were detected in the protein complex after co-IP with anti-Flag magnetic beads (*Figure 1—figure supplement 2E*), demonstrating the interaction of exogenous CD44 and CD81 in these cells.

According to predicted protein structures for CD81 and CD44 interactions, we further designed a deletion mutant of CD81d by removing the loop region of amino acids 159–187 (CD81d), which expression was stable and presented on the cell membrane (*Figure 1—figure supplement 2F*), and

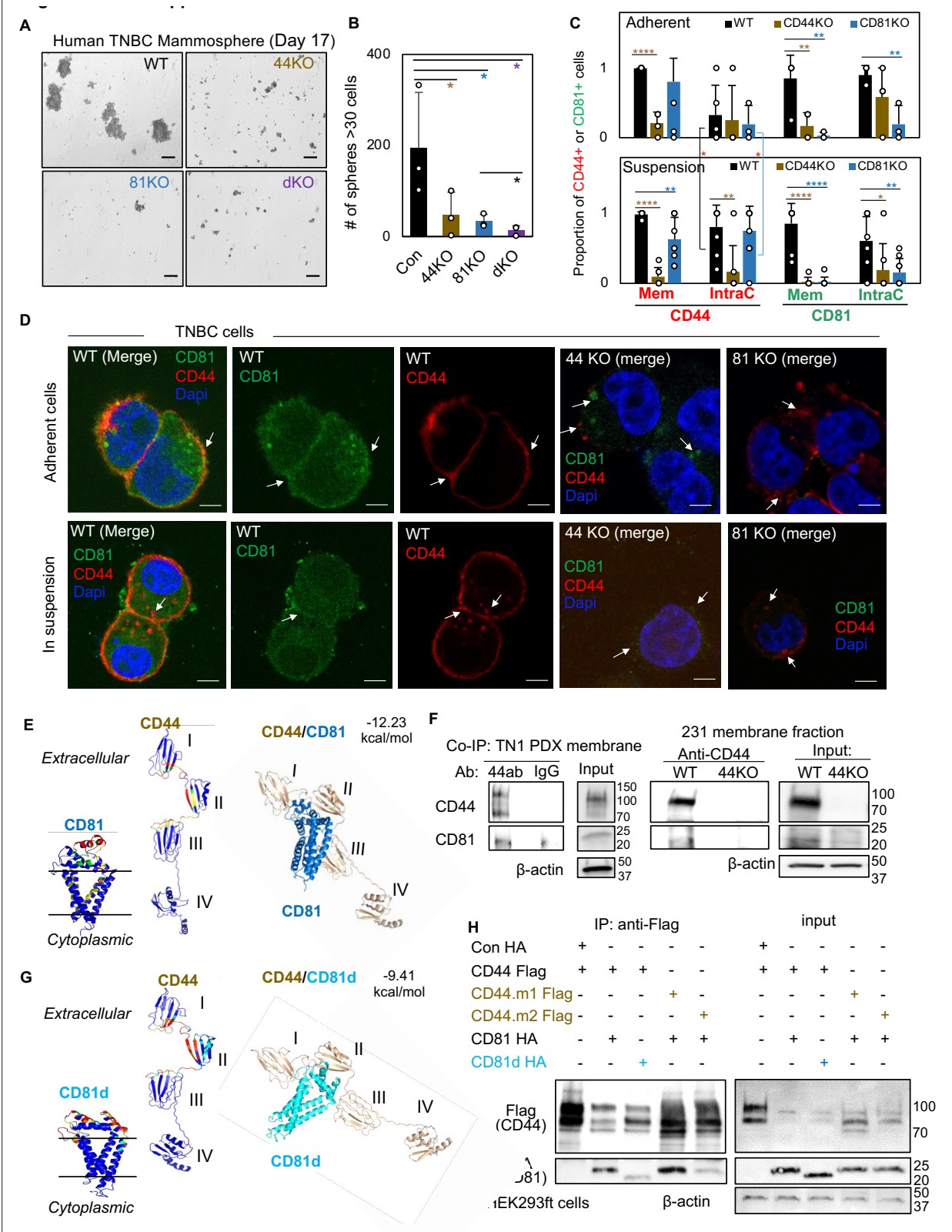

**Figure 1.** CD81 interacts with CD44 on the membrane and promotes mammosphere formation of triple negative breast cancer (TNBC) cells. Representative images (**A**) and bar graphs (**B**) of the mammospheres of MDA-MB-231 cell groups (WT, CD44KO, CD81KO, dKO pool populations), 17 days after seeded at 2000 cells per well (6-well plate) in serum-free mammosphere formation media. $N$ = 4 replicates. Error bars represent standard deviation. Repeated three times. Scale bar = 100 µm. One-tailed Student $T$-test *p < 0.05. Bar graphs (**C**) and representative images (**D**) of

*Figure 1 continued on next page*

*Figure 1 continued*

membrane and intracellular CD44 and CD81 localization in WT, CD44KO, and CD81KO MDA-MB-231 cells (adherent and in suspension), detected by immunofluorescence staining with anti-CD44-Texas Red, anti-CD81-Alexa488, and DAPI (4′, 6-diamidino-2-phenylindole) for DNA staining. Sample size $N = 20$ (WT), 14 (CD44KO), and 20 (CD81KO) adherent cells and $N = 31$ (WT), 31 (CD44KO), and 28 CD81KO cells in suspension. Error bars represent standard deviation. Scale bar = 5 µm. Two-tailed Student $T$-test p = 0.03 for intracellular CD44 levels between adherent and suspension cells (both WT and CD81KO cells). Compared to WT cells, $T$-test p values for adherent CD44KO/CD81KO cells are 9.52e−06/0.22 (membrane CD44), 0.81/0.65 (intracellular CD44), 0.009/0.006 (membrane CD81), 0.15/0.002 (intracellular CD81); and p values for cells in suspension are 1.21e−11/0.003 (membrane CD44), 0.002/0.72 (intracellular CD44), 3.08e−06/2.64e−07 (membrane CD81), and 0.04/0.004 (intracellular CD81). Significant pair comparison differences are marked in the graph. ANOVA analyses among three groups with p values for adherent and in-suspension cells, respectively: 0.0004 and 7.357E−08 (membrane CD44), 0.9119 and 0.0024 (intracellular CD44), 0.0022 and 1.079E−09 (membrane CD81), and 0.0255 and 0.0103 (intracellular CD81). (**E**) Predictive modeling of CD44 and CD81 interaction with hot spots shown in red and yellow. Top-ranked structural models of predictive interactions between CD44/CD81 (estimated binding energy: −12.23 kcal/mol) and between CD44/CD81d (deletion mutant) (estimated binding energy: −9.41 Kcal/mol). (**F**) Immunoblots of endogenous CD44 and CD81 immunoprecipitated by anti-CD44 from the lysates (membrane fraction) of WT/44KO MDA-MB-231 and (total) WT TN1 patient-derived xenograft (PDX) cells. (**G**) Predictive modeling of CD44 and CD81d (deletion mutant) interaction with hot spots shown in red and yellow. Top-ranked structural models of predictive interactions between CD44/CD81d (deletion mutant) (estimated binding energy: −9.41 kcal/mol). (**H**) Representative immunoblots of CD81-HA (CD81d) immunoprecipitated by anti-CD44-Flag (CD44 mutants CD44.1 and CD44.2) from the lysates of HEK293ft cells ($N = 3$ biological replicates; Con, control). Cells were transfected with either Flag-CD44 or mutants (Flag-CD44.1, Flag-CD44.2) and Con HA, HA-CD81, or HA-CD81d with deletion at amino acids 159–187 (CD81d). Forty-eight hours after transfection the cells were harvested for Flag IP.

The online version of this article includes the following source data and figure supplement(s) for figure 1:

**Source data 1.** Uncropped blots associated with *Figure 1F*.

**Source data 2.** Uncropped blots associated with *Figure 1H*.

**Figure supplement 1.** Characterization of CD44KO and CD81KO breast cancer cell lines.

**Figure supplement 1—source data 1.** Uncropped blots associated with *Figure 1—figure supplement 1B*.

**Figure supplement 1—source data 2.** Uncropped blots associated with *Figure 1—figure supplement 1C*.

**Figure supplement 1—source data 3.** Uncropped blots associated with *Figure 1—figure supplement 1D*.

**Figure supplement 2.** CD81 colocalizes with CD44 on cellular membrane in breast cancer cell line.

**Figure supplement 2—source data 1.** Uncropped blots associated with *Figure 1—figure supplement 2C*.

**Figure supplement 2—source data 2.** Uncropped blots associated with *Figure 1—figure supplement 2E*.

predicted to dramatically impair the interaction, with an estimated free energy of binding altered to −9.41 kcal/mol (*Figure 1G*). We then assessed the effects of CD81d on its interaction with CD44, and the effects of CD44 mutants CD44.m1 and CD44.m2, with point mutations in domains I and II (*Kawaguchi et al., 2020*), respectively, on their interaction with CD81 (*Figure 1—figure supplement 2G*). By three independent co-IP tests using the *CD44-Flag* (WT or mutant) and *CD81-HA* (WT or mutant) co-transfected cells, we found that CD81d and CD44.m2 mutant partially lost their capability to interact with CD44 and CD81, respectively (*Figure 1H*), indicating the specific regions required for CD81–CD44 interactions.

To determine if CD81 reintroduction rescues mammosphere formation, CD81KO MDA-MB-231 cells were overexpressed with CD81 (HA/GFP tagged) and assessed for mammosphere formation which yielded comparable capacity to WT cells (*Figure 1—figure supplement 2H,I*). Notably, when HA-CD81d mutant was overexpressed in CD81KO MDA-MB-231 cells, mammosphere formation capacity was significantly compromised (*Figure 1—figure supplement 2H*), suggesting CD81 binding with CD44 is important for functions of self-renewal as measured by mammosphere formation.

## Bioinformatic analyses reveal endocytosis pathway regulated by CD81 and CD44

To elucidate the membrane protein CD81-regulated molecular networks and pathways in self-renewal, we analyzed mass spectrometry-based global proteomes and phosphoproteomes as well as RNA-sequencing-based transcriptomes of TNBC cells (adherent or in suspension) with CD81 and CD44 depletion.

We first performed RNA sequencing to examine the CD81 KD effects on transcriptome in adherent MDA-MB-231 cells after si*CD81* transfections (*Figure 2—figure supplement 1A*). The Metascape pathway (*Tripathi et al., 2015*) analysis identified CD81 KD-influenced transcriptome in the pathways

of protein modification, cell proliferation, and differentiation (*Figure 2—figure supplement 1B*). However, among a few hundreds of significantly altered transcripts, most of them had very low base-line detection, and only a few genes with robust expression were upregulated over twofold (such as *SEMA7A*, *HMGA2*, *ACVR2B*, *YOD1*, and *FUT4*) or downregulated more than half, including *CDC34*, *UBE2R2*, *SLC7A11*, and *ADIRF* (*Figure 2—figure supplement 1C*, *Supplementary file 2*, and *Supplementary file 3*). Among the si*CD81*-upregulated genes, *SEMA7A*, a glycosylphosphatidylinositol membrane anchor promoting osteoclast and blood cell differentiation (*Delorme et al., 2005*; *Jaimes et al., 2012*), was depleted in CD81KO cells and si*SEMA7a* partially rescued or restored mammo-sphere formation in these cells (*Figure 2—figure supplement 1D,F*), suggesting a role of SEMA7A in inhibiting self-renewal of breast cancer cells.

To further explore the possible proteome alterations connecting CD81 with CD44 regulation in TNBC cells, we pursued mass spectrometry analyses of these cells with transient KDs. Our pilot proteomic comparisons between adherent versus clustering MDA-MB-231 cells in-suspension, showed minimal protein level alterations (five proteins with >twofold changes) but drastic phosphoproteomic alterations (over 1300 phosphopeptides with >twofold changes) (*Figure 2—figure supplement 2A*, *Supplementary file 4*), suggesting that posttranslational phosphorylation significantly modulates the signaling pathways of cells in suspension that mimick detached migrating cells and CTCs.

We then collected three sets of control, si*CD81*- and si*CD44*-transfected cells in suspension (3 hr clustering) with a transient knockdown of CD81 or CD44 (*Figure 2—figure supplement 2B*) for both global proteome and phosphoproteome analyses. By comparative mass spectrometry analyses of three groups of cell lysates, we identified six clusters of altered general proteins (G1–G6) and another six clusters of altered phosphoproteins (P1–P6) with top pathways for each cluster annotated by KEGG (Kyoto Encyclopedia of Genes and Genomes) (*Ogata et al., 1999*; *Figure 2A–D*, *Supplementary file 5*, *Supplementary file 6*, and *Supplementary file 7*). In comparison to the control cell profiles, Clusters G1 (downregulated) and G6 (upregulated) represent the altered proteins in both si*CD81* and si*CD44* cells in the same directions, suggesting shared pathway regulation, such as DNA replication and cell cycle in G1 and metabolic pathways in G6, involved in the regulation of self-renewal and cell proliferation. Ribosome pathway was the top pathway in Clusters G3 and G4 with downregulated proteins specific to si*CD44* KD and si*CD81* KD, respectively, suggesting distinct pathway components but possible contribution to similar ribosome-regulating functions of CD81 and CD44 in TNBC cells.

Most notably, endocytosis, lysosome, and proteosome pathways became part of the top signature components of G2, G4, and G5 clusters distinctly altered in si*CD81*- and si*CD44*-transfected cells (*Figure 2A,B*), indicating one of the central pathways distinctly regulated by CD44 (RAB4A, RAB11A, RAB11B, SNX12, SNX4, and SNX6) and by CD81 (VPS29, SNX3, SNX1, SNX2, CAV2, and RAB7A) (*Supplementary file 5* and *Supplementary file 6*). It is well known that exosomes are generated through the endocytic pathway connected with lysosomes and exocytosis.

Interestingly, the alterations in endocytosis pathway components became more evident when phosphoproteome data were analyzed among three groups of cell lysates. Endocytosis was the top altered pathway within four out of six phosphoproteome clusters (P2, P3, P5, and P6), covering both shared and distinct signaling components altered by si*CD44* and si*CD81*. For example, both KDs promoted phosphorylation in the same residues of RAB11FP1, RAB11FP5, EPN2, and SNX12; si*CD44* specifically downregulated phosphorylation in RAB8A and PLD2 but upregulated phosphorylation in GBF1, VPS26A, SNX4, RAB11FIP1, RAB11FIP5; whereas si*CD81* downregulated phosphorylation in CAV2, ARFGAP1, and RAB11FIP5 (*Figure 2C,D*, *Figure 2—figure supplement 2C,D*, *Supplementary file 5*, and *Supplementary file 7*). These phosphoproteins regulate membrane trafficking, endosomal recycling, and caveolar formation, revealing previously unreported signaling components shared and distinguished between CD44 and CD81.

By combining 11 algorithms for machine learning-based kinase prediction, we identified the top candidates of upstream kinases potentially responsible for si*CD81*- and si*CD44*-induced alterations in phosphoproteome clusters (P1–P6), especially shared Cluster 4 with decreased phosphopep-tides catalyzed by CDK1, CDK2, and EGFR, and Cluster P3 with upregulated phosphorylation by CSNK2A1 (Casein Kinase 2 α1 subunit) in (*Figure 2—figure supplement 2E*). Furthermore, the kinase reactome analyses identified the phosphoproteome kinase networks shared between CD81 and CD44 regulation, including PAK2, one of our previously identified targets of CD44 in tumor cluster formation (*Liu et al., 2019*; *Figure 2—figure supplement 3A,B*, *Supplementary file 7*).

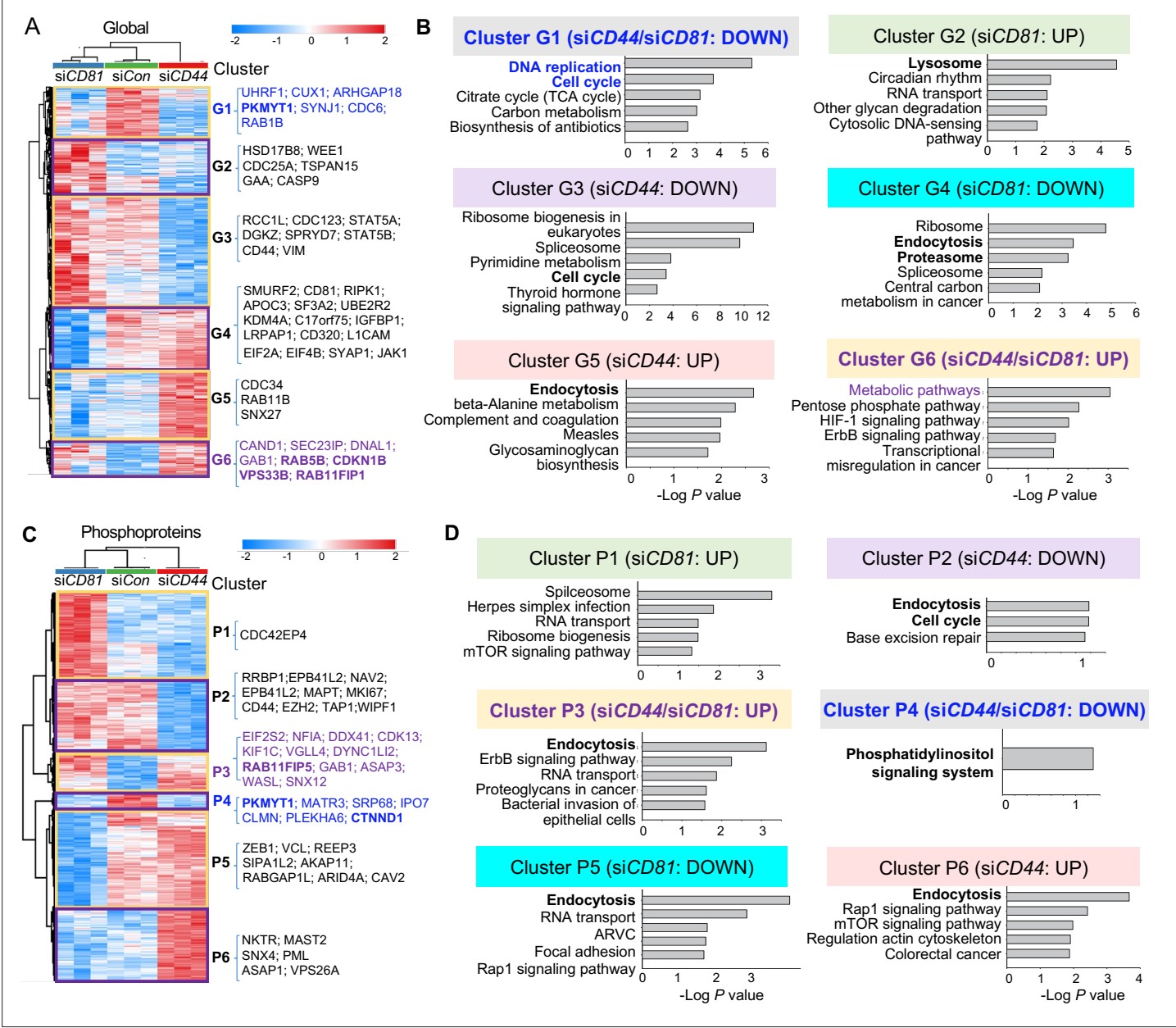

**Figure 2.** Global mass spectrometry and phosphoproteomic profiling of MDA-MB-231 cells with si*CD81* and si*CD44* KDs. Global mass spectrometry heatmap (**A**) and KEGG pathway analysis (**B**) of altered top pathways with significantly expressed proteome within six different clusters (G1–G6) in siControl, si*CD81*, and si*CD44* cells at 3 hr in suspension (*N* = 3 replicates, ANOVA *T*-test false discovery rate (FDR) <0.01, p = <0.05). Phosphoproteomic mass spectrometry heatmap (**C**) and KEGG pathway analysis (**D**) of altered top pathways with significantly changed phosphoproteome within six different clusters (P1–P6) in siControl, si*CD81*, and si*CD44* cells at 3 hr in suspension (*N* = 3 replicates, ANOVA *T*-test FDR <0.01, p = <0.05).

The online version of this article includes the following source data and figure supplement(s) for figure 2:

**Figure supplement 1.** SEMA7A increased in CD81-depleted cells.

**Figure supplement 1—source data 1.** Uncropped blots associated with *Figure 2—figure supplement 1C*.

**Figure supplement 1—source data 2.** Uncropped blots associated with *Figure 2—figure supplement 1F*.

**Figure supplement 2.** Global and phosphoproteomic analyses in CD81- and CD44-depleted cells.

**Figure supplement 2—source data 1.** Uncropped blots associated with *Figure 2—figure supplement 2B*.

**Figure supplement 3.** The kinase reactome networks in CD81- and CD44-depleted cells.

**Figure supplement 3—source data 1.** Uncropped blots associated with *Figure 2—figure supplement 3C*.

These data are consistent with our previous discoveries on EGFR and PAK2 which are two representative components of the CD44 signaling and functional regulation in tumor cell clustering and metastasis (*Liu et al., 2019*; *Liu et al., 2021*). RAB11FIP1, RAB7A, RAB8A, and RAB10 proteins were also validated via immunoblot analysis as targets from the proteomic screens and confirmed at elevated levels in CD81KO and CD44KO MDA-MB-231 cells compared to WT cells (*Figure 2—figure supplement 3C*).

From an independent proteome study using the LC/MS/MS profiles of adherent MDA-MB-231 cells, the GO Processes analysis of upregulated proteins in CD44KO cells indicates regulation of proteolysis and exocytosis (*Figure 3—figure supplement 1A*, *Supplementary file 8*), whereas the GO Localization analysis of the downregulated proteins in CD44KO cells revealed the top hits of altered proteins in extracellular exosome/vesicle/organelle, previously unknown to be associated with CD44 functions (*Figure 3—figure supplement 1B*), validating the phosphoproteome pathway regulations by CD44 and CD81 in membrane trafficking, endocytosis, lysosomes, and exocytosis in *Figure 2*. Thus, we hypothesized that CD44 and CD81 regulate EV/exosome biogenesis production.

## CD44 and CD81 are required for EV integrity and EV-promoted mammosphere formation

CD81 is one of the most classical markers enriched in large and small EVs (*Kowal et al., 2016*; *Mathieu et al., 2021*; *Bobrie et al., 2011*; *Théry et al., 2006*); however, its functions in EVs are relatively unknown. Based on our findings that CD81 and CD44 share many signaling components regulating endocytosis and membrane trafficking, we continued to investigate whether CD44 or CD81 regulates EV biogenesis and/or functions.

Using ultra-high resolution transmission electron microscopy, we discovered enlarged multivesicular bodies (accumulated endosomes), increased vacuoles (early endosomes or endocytic vesicles), and altered lysosomes in both CD44KO and CD81KO cells in comparison to WT control of MDA-MB-231 cells (*Figure 3A,B*, *Figure 3—figure supplement 1C*). CD81KO and CD44KO cells secreted a higher number of EVs (particles per cell) to the culture supernatants than the WT cells, as directly measured by the microflow vesiclometry (MFV) (*El-Shennawy et al., 2020*; *Kibria et al., 2016*) or by nanoparticle tracking analysis (NTA) on NanoSight after EV purification via 100,000 × *g* ultracentrifugation (*Figure 3C*, *Figure 3—figure supplement 1D,E*). The purified EVs (ev44KO and ev81KO) were deficient for both CD44 and CD81 while they showed similar sizes to the WT control with similar EV protein markers such as TSG101 and LAMP2b (*Figure 3D–F*). However, when examined by cryo-EM, ev81KO displayed impaired membrane integrity (*Figure 3G,H*), indicating an essential role for CD81 in modulating EV biogenesis and packaging of membrane proteins.

A mass spectrometry analysis of the ev44WT and ev44KO identified 26 out of 416 exosomal proteins differentially expressed in ev44KO, including relatively decreased CD81 and syntenin-1 as well as upregulation of RAB-11B in association with exosome biogenesis (*Supplementary file 9*). Proteomic pathway and network analyses identified the top altered signaling pathways related to cell adhesion, integrin-mediated matrix adhesion, cell cycle, and cytoskeleton regulation and rearrangement, and a network linking to integrins and focal adhesion (*Figure 3—figure supplement 2A–F*), which may contribute to CD44-mediated functions in tumor metastasis (*Liu et al., 2019*; *Liu et al., 2021*).

Next, we investigated whether, upon cellular uptake, cancer exosomes could rescue any self-renewal defects of CD81KO recipient cells, such as mammosphere formation. To do this, CD81KO MDA-MB-231 cells were educated with phosphate-buffered saline (PBS) or exosome-enriched EVs (evWT, ev44KO, and ev81KO) for 2 weeks (*Figure 3I*). Following education, the cells were evaluated for self-renewal related properties. The cells educated by evWT formed larger mammospheres with elevated protein levels of OCT4, pSTAT3, and FAK than the cells treated with PBS, ev44KO, or ev81KO (*Figure 3J,K*, *Figure 3—figure supplement 3A*), demonstrating that EV-presented proteins CD44 and CD81 are required to promote self-renewal of recipient cells. Consistently, the Cd81KO 4T1 cells restored mammosphere formation after exosome education with evWT whereas no rescue effects were observed from ev81KO isolated from 4T1 cells (*Figure 3—figure supplement 3B,C*), demonstrating Cd81-dependent regulation of self-renewal induced by mouse TNBC-secreted EVs. These data might indicate a non-cell autonomous mechanism of CD44- and CD81-promoted self-renewal.

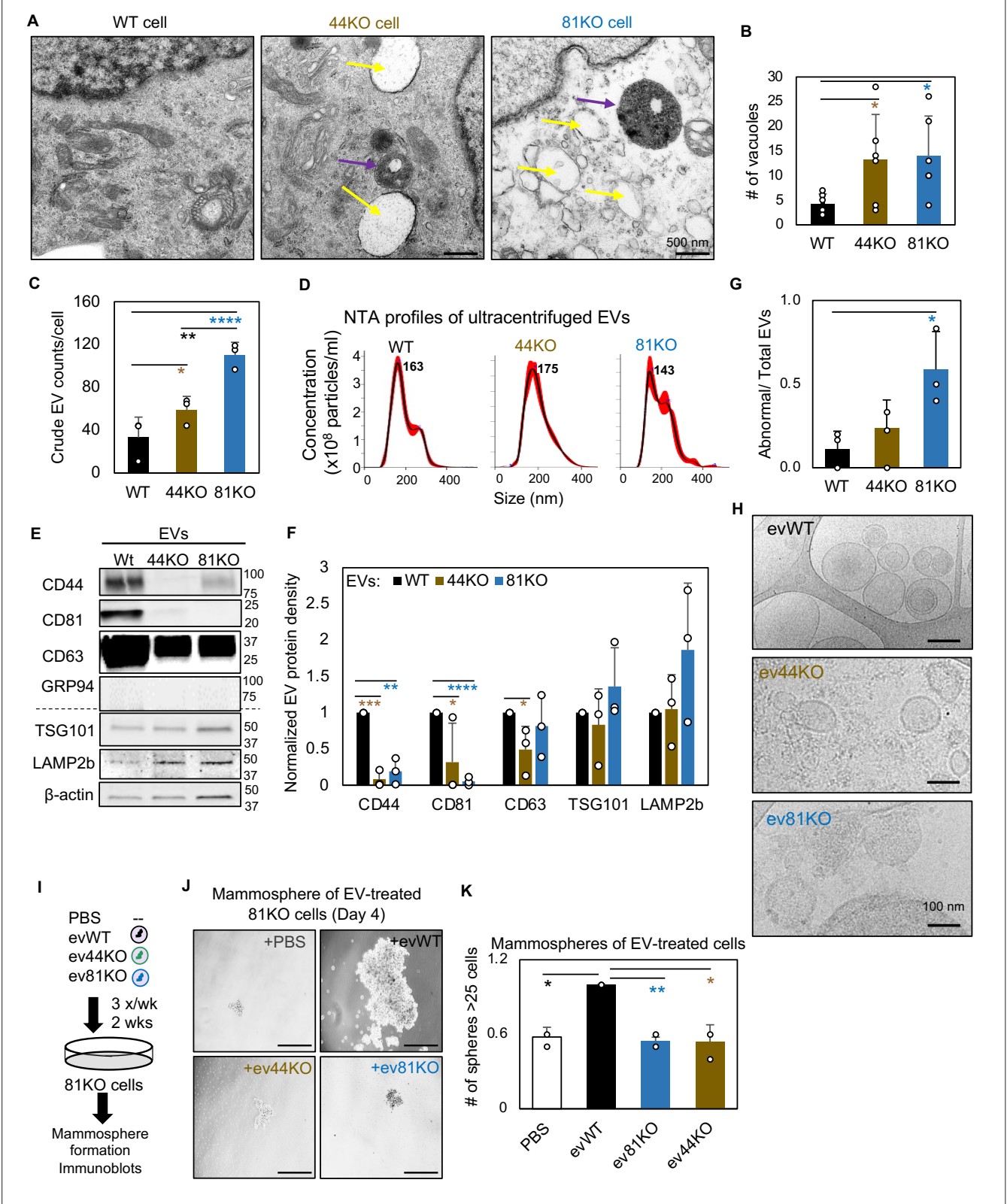

**Figure 3.** CD81 and CD44 are required for exosome-induced cancer stemness. Transmission electron microscopy images (**A**) of WT, 44KO, and 81KO MDA-MB-231 cells and number of vacuoles observed per cell (**B**). Yellow arrows point to vacuoles or early endosomes/endocytic vesicles, purple arrows to multivesicular bodies. Scale bar = 500 nm. *N*= 6. Error bars represent standard deviation. Student two-tailed *T*-test *p = 0.017. (**C**) Counts of extracellular vesicles (EVs) per cell in crude culture supernatants of WT, 44KO, and 81KO cells, measured by Apogee (*N* = 3 or 5). Error bars represent

*Figure 3 continued on next page*

*Figure 3 continued*

standard deviation. Two-tailed Student *T*-test ****p = 0.0001, **p = 0.003, one-tailed Student *T*-test *p = 0.036. Nanoparticle tracking analysis (NTA)-based size distributions (repeated three times) (**D**) and (**E, F**) representative immunoblots (*n* = 3) and quantification for EV proteins in ultracentrifuge-isolated EV particles from the culture media of WT, 44KO, and 81KO MDA-MB-231 cells. *N*= 3. Error bars represent standard deviation. *T*-test *p = 0.04/0.02 (one tailed), **p = 0.001 (two tailed), ***p = 0.0002 (two tailed), ****p = 8.5e−6 (two tailed). Cryo-EM images (repeated twice) (**G**) and quantification (**H**) of membrane integrity in evWT, ev44KO, and ev81KO, taken at ×8000 nominal magnification (a pixel size of 4.125 Å). Vesicles assessed included 18 WT, 17 CD44KO, and 63 CD81KO. Scale bar = 200 nm. (**I**) Schematic of 81KO MDA-MB-231 cells educated with phosphate-buffered saline (PBS) or evWT, ev44KO, and ev81KO every 2 days for 2 weeks and seeded at low density to evaluate mammosphere formation. Representative images (**J**) and bar graph quantification (**K**) of mammospheres 4 days after seeding of 1000 cells per well (24-well plate) after education with PBS, WT EVs, CD44KO EVs, and CD81KO EVs. Scale bar = 100 μm. *N*= 4. Error bars represent standard deviation. One-tailed Student *T*-test *p = 0.02, **p = 0.01. Repeated twice times.

The online version of this article includes the following source data and figure supplement(s) for figure 3:

**Source data 1.** Uncropped blots associated with *Figure 3E*.

**Figure supplement 1.** Effects of CD44/CD81 depletion on cellular pathways related to extracellular vesicles (EVs).

**Figure supplement 2.** Proteomic pathway analysis of evCD44 derived from CD44KO cells.

**Figure supplement 3.** CD81 is required for exosome-induced effects on stemness phenotype.

**Figure supplement 3—source data 1.** Uncropped blots associated with *Figure 3—figure supplement 3*.

## CD81 is enriched in human CTCs and promotes CTC cluster formation

We further explored the functions of CD81 in metastasis which requires self-renewal. Our previous work demonstrated that CD44 mediates tumor cell aggregation and CTC cluster formation that promotes stemness and metastasis (*Liu et al., 2019*; *Liu et al., 2021*), and is associated with reduced progression-free survival (*Mu et al., 2015*; *Aceto et al., 2014*). To determine if CD81 regulates breast cancer metastasis like CD44, we assessed the clinical relevance of CD81 expression in human breast tumors and CTCs. Public database analyses revealed that high expression of CD81 protein in breast tumors was associated with an unfavorable overall survival, relapse-free survival, and distant metastasis-free survival in patients with TNBC (Liu_2014 cohort) (*Osz et al., 2021*; *Figure 4A–C*, *Figure 4—figure supplement 1*). We also conducted a primary breast tumor tissue microarray (TMA) study and observed a correlation of CD44 and CD81 expression across tumor subtypes (TNBC, HER2, luminal A/B) as well as an upregulated expression of CD44/CD81 in TNBC in comparison to luminal A/B (*Figure 4—figure supplement 2A–D*).

We then utilized three methods, immunofluorescence staining via CellSearch, flow cytometry, and RNA sequencing data analysis to further examine the CD81 expression in the CTCs isolated from patients with breast cancer. First, the FDA-approved CellSearch platform was employed to enrich EpCAM⁺ cells via anti-EpCAM beads and conduct immunofluorescence staining for validation of CD45⁻cytokerin⁺DAPI⁺ CTCs. CellSearch-based analyses of patient blood samples revealed CD81 expression in over 90% of CTC events (*N* = 6 patients with 381 CTC events) (*Figure 4D,E*) and 100% of CTC clusters (*N* = two patients with 10 clusters). To expand the CD81 analysis in EpCAM$^{+/-}$ putative CTCs, we established a flow cytometry approach to gate single cells and clusters based on size channels (forward scatter and side scatter) as validated on clustering WT and non-clustering CD44KO tumor cells (*Figure 4—figure supplement 3A*). Consistently, flow cytometry-based analyses of putative CTCs (lineage⁻CD45⁻EpCAM$^{+/-}$) and CTC clusters showed a significant higher expression of CD81 and CD81/CD44 double-positive expression on the clusters compared to single cells (*N* = 50 patients, p = 0.0005) (*Figure 4F–H*, *Figure 4—figure supplement 3B*), similar to the CD44 expression enriched in CTC clusters (*Figure 4—figure supplement 3C*; *Liu et al., 2019*), suggesting a possible positive feedback loop between CD44 and CD81. Using publicly available datasets on single-cell RNA sequencing of Parsotix-filtered CTCs from the blood of patients with breast cancer (*Gkountela et al., 2019*; *Aceto et al., 2014*), we also found elevated CD81 expression in CTC clusters compared to single CTCs, confirming the clinical relevance of CD81 as a possible biomarker in CTC clusters and breast cancer metastasis (*Figure 4—figure supplement 3D*).

To further determine the role of CD81 in CTC clustering, we utilized TN1 and TN2 PDX tumor cells, MDA-MB-231 cells, and 4T1 cells to analyze the phenotypic changes caused by si*CD81*/si*Cd81*-mediated downregulation or CRISPR-Cas9-mediated gene depletion. CD81 depletion or downregulation resulted in compromised cluster formation in all four tested human and mouse TNBC models

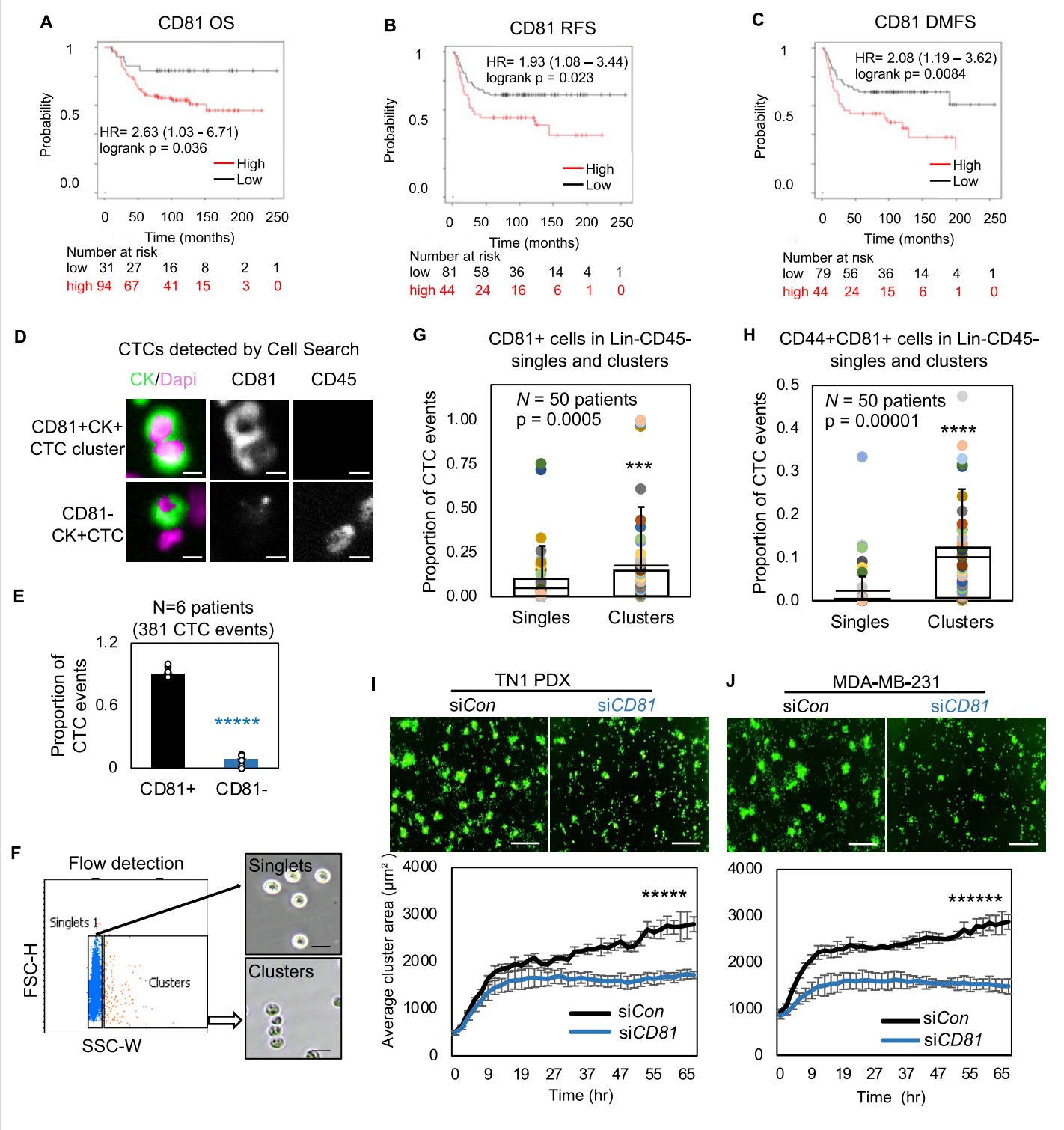

**Figure 4.** CD81 is associated with patient survival and enriched in circulating tumor cells (CTCs) promoting tumor cell aggregation. Kaplan–Meier plots of CD81 protein expression in patients with triple negative breast cancer (TNBC) correlate with an unfavorable overall survival (**A**), relapse-free survival (**B**), and distant-metastasis-free survival (**C**). Representative images (**D**) and quantified % (**E**) of CD81+ and CD81− CTC events in the blood of six patients with metastatic breast cancer, analyzed on CellSearch. Scale bar = 5 µm. *N*= 3. Error bars represent standard deviation. Two-tailed Student *T*-test *****p = 9E−11. Representative images of flow cytometry gated singlets and clusters (**F**, scale bar = 25 µm) and bar graphs of proportion of CD81+ (**G**) and CD44+CD81+ (**H**) in putative Lin-CD45- CTC events (622,509) in the blood (*N* = 50 patients). *N*= 50. Error bars represent standard deviation. Two-tailed Student *T*-test ***p = 0.005, ****p = 0.00001. IncuCyte images (top panels) and quantified tumor cell aggregation curves of TN1 patient-derived

*Figure 4 continued on next page*

*Figure 4 continued*

xenograft (PDX) (**I**), MDA-MB-231 (**J**) cells upon CD81 KD (repeated at least three times). Scale bar = 300 µm. *N*= 5. Error bars represent standard deviation. Two-tailed Student *T*-test *****p = 1E−12, ******p = 8E−19.

The online version of this article includes the following figure supplement(s) for figure 4:

**Figure supplement 1.** Kaplan–Meier plots of patient survival based on CD81 protein expression.

**Figure supplement 2.** Tissue microarray (TMA) clinical characteristics.

**Figure supplement 3.** Gating strategy of circulating tumor cells (CTCs) by flow cytometry and the RNA expression level of CD81 in single versus clustered CTCs.

**Figure supplement 4.** Cluster formation of tumor cells (MDA-MB-231, 4T1, and patient-derived xenograft [PDX] models) in CD81 and CD44 modified cells.

**Figure supplement 5.** CD81 KD inhibits breast cancer cell migration and invasion.

(*Figure 4I,J*, *Figure 4—figure supplement 4A–C*), suggesting an important role of CD81 in promoting tumor cell aggregation. To determine if CD81 reintroduction rescues tumor cluster formation, we also assessed the clustering efficiency of CD81KO MDA-MB-231 cells when overexpressed with CD81 (HA tagged) which restored tumor clustering close to WT cells, whereas HACD81d mutant failed to do so (*Figure 4—figure supplement 4D*), suggesting CD81 binding with CD44 is important for functions of tumor cluster formation.

Consistently, an anti-CD81 activating agonist promoted breast tumor cell clustering in a CD81- and CD44-dependent manner as the cluster-enhancing effects of the antibody diminished in both CD81KO and CD44KO cells (*Figure 4—figure supplement 4E*), further demonstrating the interplay and cross-dependence between CD81 and CD44 for optimal cluster formation. In addition, we performed a scratch wound assay with and without Matrigel coverage to evaluate cell invasion and migration, respectively. While CD81KO tumor cells closed the wound at a slightly slower migration speed in the absence of Matrigel compared to that of the WT control, both CD81KO cells and CD44KO cells showed much more similar and more dramatic reduction in cell invasion (*Figure 4—figure supplement 5A,B*). We proposed to determine the role of CD81 in tumorigenesis and metastasis in the next experiment.

## CD81 promotes tumorigenesis and lung metastasis of TNBC

We first examined the importance of human CD81 and mouse Cd81 in tumorigenesis of TNBC. After dissociation, CD81$^+$ and CD81$^-$ TN1 PDX tumor cells were sorted on a fluorescence-activated cell sorter and then injected at dilutions of 1000 and 100 cells per injection into the mammary fat pads of NSG mice (*n* = 4 injections/group). Up to 45 days after injection, we observed compromised tumor initiation and growth in the CD81$^-$ cells, especially at the 100 cell dilution, compared to the CD81$^+$ cells that grew tumors at both dilutions (*Figure 5A–C*). Similar tumorigenesis data were observed in Cd81KO 4T1 cells following 1000 and 100 cell injections (*Figure 5D,E*) (*n* = 8 injections/group). Furthermore, CD81KO MDA-MB-231 cells had compromised tumor growth (weight) and tumorigenesis compared to WT controls when orthotopically implanted at 100 and 10 cells per mouse mammary fat pad injection (*Figure 5F,G*) (*n* = 8 injections/group). When CD81 was stably overexpressed in CD81KO MDA-MB-231 cells tumorigenesis and tumor growth was rescued (*Figure 5F,G*). These data demonstrate that CD81 is a newly identified promoter of breast tumor initiation.

We continued to determine if CD81 drives spontaneous lung metastasis in vivo. Considering a slightly decreased tumor growth rate in mouse Cd81KO 4T1 cells, we implanted 1000 WT cells and 6000 Cd81KO 4T1 cells into the mammary fat pads of Balb/c mice to achieve comparable tumor burden on day 52 when tumors and lungs were harvested (*Figure 6A,B*) (*n* = 5 mice/group; 2 injections/mouse). While there were no significant differences in breast tumor weight between the two groups, Cd81KO tumor cells failed to metastasize to the lungs, with a significantly lower number of metastatic colonies than those of WT tumors (*Figure 6A–C*). Furthermore, the mice bearing Cd81KO tumors had 100% survival compared to 0% survival of the mice with oversized WT tumors by day 52 (*Figure 6D*).

Furthermore, when human TNBC cells were implanted orthotopically at 10,000 cells to ensure tumor growth, CD81KO cells phenocopied CD44KO cells with impaired or lost capability to develop spontaneous lung metastases in mice after being normalized by tumor burden (*Figure 6E–H*) (*n* =

| Cells | No. of cells | CD81+ (CD81WT/ Cd81WT) | CD81- (CD81KO/ Cd81KO) | CD81KO +81OE | p value 81+/WT v 81-/KO | p value 81+/WT v 81OE | p value 81-/KO v 81OE |
|---|---|---|---|---|---|---|---|
| TN1 PDX | 1,000 | 10/10 | 8/10 | NA | 0.075 | NA | NA |
| | 100 | 12/16 | 6/16 | NA | 0.016 | NA | NA |
| 4T1 (WT/KO) | 1,000 | 8/8 | 6/8 | NA | 0.07 | NA | NA |
| | 100 | 8/8 | 1/8 | NA | <0.0001 | NA | NA |
| MDA-MB-231 (WT/KO/OE) | 100 | 20/20 | 14/20 | 8/8 | 0.0035 | NS | 0.03 |
| | 10 | 8/8 | 5/8 | 6/8 | 0.029 | 0.074 | 0.31 |

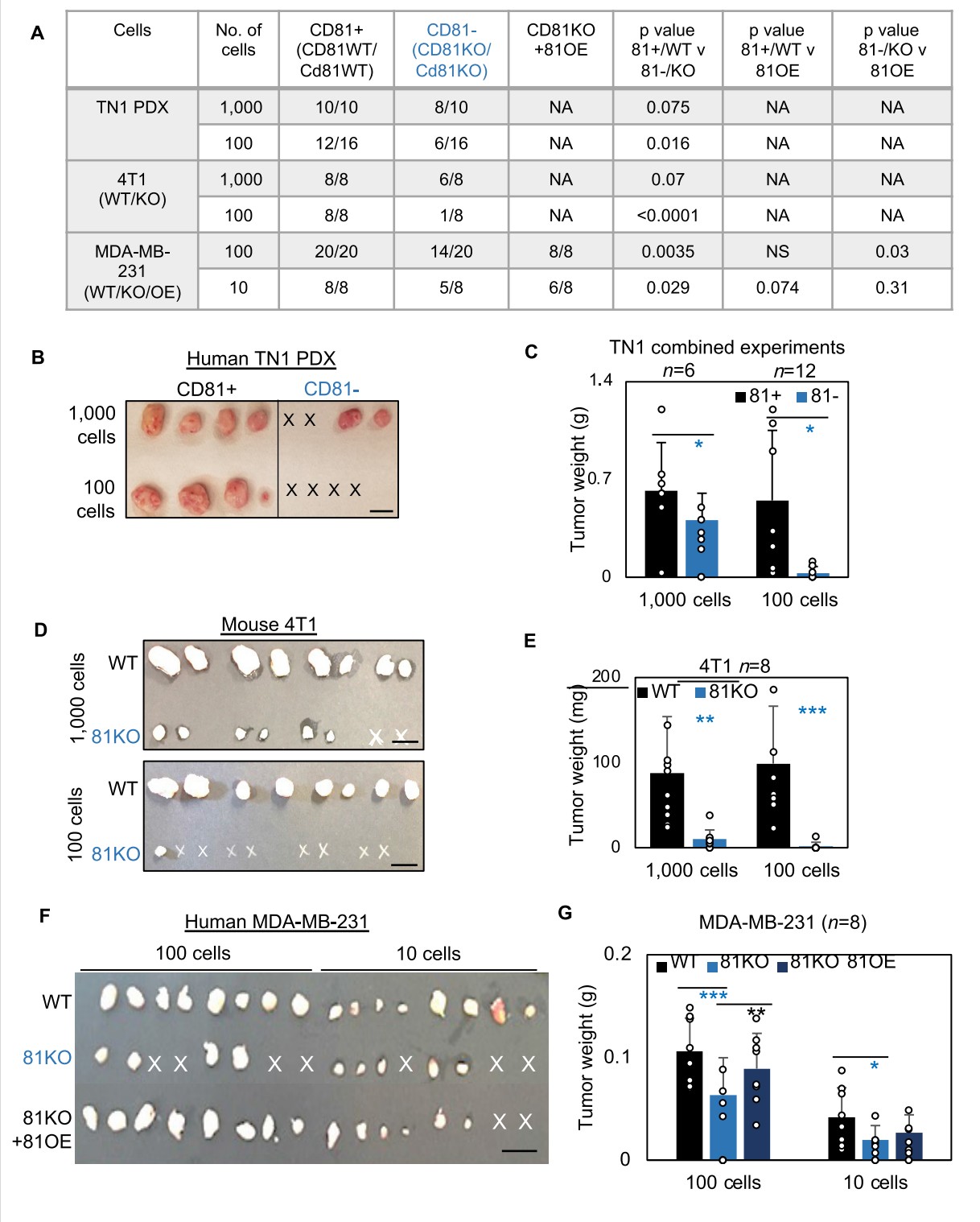

**Figure 5.** CD81 promotes tumorigenesis of triple negative breast cancer (TNBC) cells. (**A**) Table of serial dilutions of tumorigenic results with CD81⁺ and CD81⁻TN1 PDX, CD81 WT/KO 4T1 cells, and CD81 WT/KO/KO + CD81 overexpression (OE) MDA-MB-231. One-tailed Student *T*-test. Pictures of harvested tumors (**B**) and graphs of tumor weight comparisons (**C**) with CD81⁺ and CD81⁻TN1 PDX tumor implants. Scale bar = 1.3 cm. *n* = 10–16 injections per group. *N*= 6-12. Error bars represent standard deviation. Two-tailed Student *T*-test comparisons between CD81⁺ and CD81⁻ cells: *p = 0.026 (1000 cells), 0.019 (100 cells). Pictures of harvested tumors (**D**) graphs of tumor weight comparisons (**E**). Scale bar = 1 cm. *n* = 8 injections (2 injections/mouse). *N*= 8. Error bars represent standard deviation. Two-tailed Student *T*-test **p = 0.002, ***p = 0.001. Pictures of harvested tumors (**F**)

*Figure 5 continued on next page*

*Figure 5 continued*

and graphs of tumor weight comparisons (**G**) with CD81 WT/KO/KO + CD81 OE MDA-MB-231 cells. Scale bar = 1 cm. *n* = 8 injections (4 injections/ mouse). *N*= 8. Error bars represent standard deviation. Two-tailed Student *T*-test ***p = 0.0007, **p = 0.006, *p = 0.026.

5 mice/group; 2 injections/mouse), resulting in a significantly better survival of the mice bearing KO tumors than those bearing WT tumors (**Figure 6I**). Consistently, diminished lung colonization or experimental metastasis were observed in the NSG mice 26 days after receiving CD81KO cells in comparison with that of WT cells via tail vein injection, as measured via bioluminescence imaging, fluorescence microscopy, and HE-staining of lung tissues (**Figure 6J–L**, **Figure 6—figure supplement 1A**) (*n* = 3 mice/group). Meanwhile, siCD81 transfection-mediated transient KD of CD81 also reduced the metastatic potential of MDA-MB-231 cells following tail vein injection (**Figure 6—figure supplement 1B,C**) (*n* = 3 mice/group).

Finally, we measured the CTC events in the blood within 1 day following tail vein injection of 4T1 tumor cells and found that Cd81 KO cells (singles and clusters) were less detectable than the WT controls, in parallel with reduced seeding for lung colonization (**Figure 6M–O**) (*n* = 3 mice/group).

In a summary, CD81 is a novel partner interacting with CD44 in breast tumor-initiating cells and promotes exosome-induced self-renewal (mammosphere formation and signature markers), tumor cluster formation, and therefore enhancing tumor initiation and lung metastasis of TNBC with an unfavorable overall survival and metastasis-free survival (**Figure 6—figure supplement 2**).

## Discussion

Utilizing machine learning-assisted experimental tests, our study reports a new role of CD81 as a binding partner of CD44 in self-renewal related mammosphere formation, quality control of EV biogenesis, EV-enhanced self-renewal of recipient cells, CTC cluster formation, and lung metastasis in TNBC in close association with clinical outcomes. While CD81 function has primarily been studied in immune cells (**Vences-Catalán et al., 2015**; **Maecker and Levy, 1997**; **Quast et al., 2011**; **van Zelm et al., 2010**), this newly identified function of CD81 in cancer metastasis is tumor cell intrinsic and can be independent of adaptive immunity, as shown in both immunocompetent and immunocompromised mice. While deep learning (**Baek and Baker, 2022**) and recent release of 200 million protein structures predicted by AlphaFold (https://alphafold.ebi.ac.uk) have transformed the structural biology and expedited the protein discovery process, machine learning platforms remain complementary and demanding to facilitate individual protein networking and interaction studies.

Tetraspanin proteins, such as CD81, are best known for making up tetraspanin enriched microdomains (**Zöller, 2009**; **Le Naour et al., 2006**). These microdomains are crucial in regulating the motility and interactions of cancer cells with their microenvironment by organizing other transmembrane proteins, such as cell adhesion molecules, growth factors, and proteases (**Zöller, 2009**; **Le Naour et al., 2006**). A study by Perez-Hernandez et al. also observed CD44 among the EV protein interactome network pulled down by CD81 peptides without exploring their relevance to EV functions (**Perez-Hernandez et al., 2013**). Our mass spectrometry-based proteomic and phosphoproteomic profiles provide comprehensive analyses of shared and distinct signaling pathways related to CD81 and CD44 functions, including cell cycle, proliferation, and metabolism in addition to EV-related endocytosis and exocytosis. Nevertheless, other tetraspanin proteins, such as TSPAN8 and CD9, may promote cancer self-renewal in colorectal cancer (**Zhu et al., 2019**) and glioblastoma (**Podergajs et al., 2016**), respectively. A potential anti-CD81 therapeutic strategy was identified to block the prometastatic effect of CD81 in animal studies (**Vences-Catalán et al., 2021**). CD44 can also interact with other transmembrane proteins such as TM4SF5, resulting in elevated properties of self-renewal and circulating capacity in hepatocarcinoma cells (**Lee et al., 2015**).

CD44 is a multifunctional class I transmembrane glycoprotein and is widely used as a marker of breast tumor-initiating cells, especially in TNBC (**Idowu et al., 2012**). While our previous work demonstrated that CD44 homophilic binding mediates tumor cell aggregation (**Liu et al., 2019**; **Kawaguchi et al., 2020**), in this study we propose that CD81 interacts with CD44 to promote intracellular CD44-CD81 heterodimer formation and possibly intercellular tetramer formation between two neighboring cells. CD44 and CD81 interactions on the cell membrane might provide feedback for protein networks and modifications. Notably, a transmembrane ubiquitin ligase family member MARCH8 has been

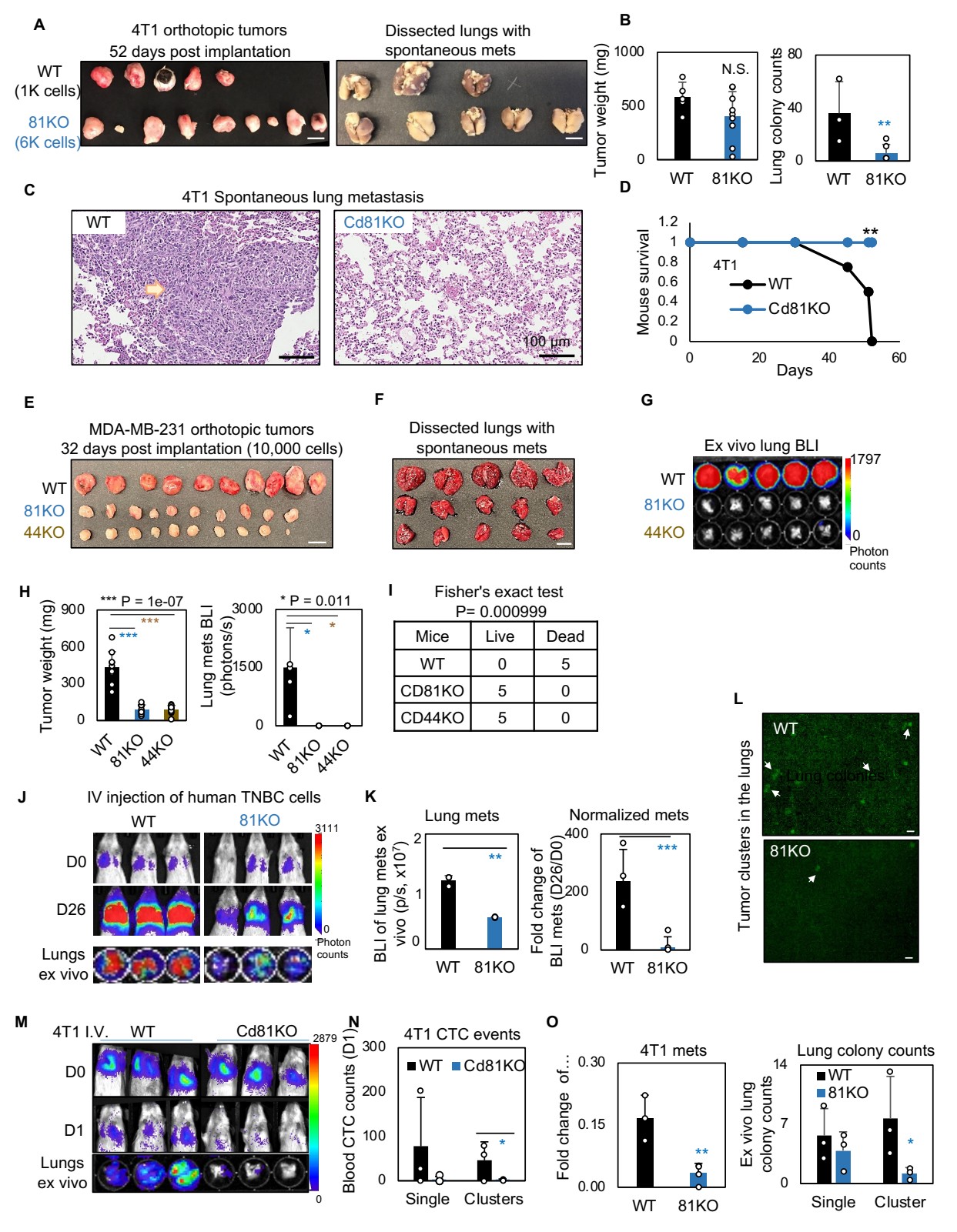

**Figure 6.** CD81 deficiency abrogates lung metastasis in breast cancer. (**A**) Photos of 4T1 orthotopic tumors (left panels) grown from implantations of comparable (1000) WT and (6000) Cd81KO cells into the L4/R4 mammary fat pads (N = 5 Balb/c mice with 10 injections) and the fixed lungs with overt metastatic colonies. By the terminal day 52, two mice from the WT group died and the left three were sick and sacrificed. Scale bar = 1 cm. (**B**) Bar graph of the tumor weights and lung colonies count between the WT tumors and KO tumors. (**C**) IHC (HE) images of lung colonies from the WT group mice

*Figure 6 continued on next page*

*Figure 6 continued*

(a higher number of visible lung metastases at a larger size) as compared to the Cd81 KO group. Scale bar = 100 µm. N.S. = not significant, two-tailed Student *T*-test **p = 0.01 (*n* = 5 mice). Error bars represent standard deviation. . (**D**) Distinct mouse survival between 4T1 WT and Cd81 KO tumor bearing mice with spontaneous lung metastases. Two-tailed Student *T*-test **p = 0.01 (*n* = 5 mice). Photos of WT and 81KO MDA-MB-231 orthotopic tumors (**E**) grown from 10,000 cell implantations and dissected mouse lungs on day 32 (**F**) (*N* = 5 NSG mice with 10 injections) Scale bar = 1 cm. Error bars represent standard deviation. (**G**) BLI images and of spontaneous metastases in the lungs ex vivo following after orthotopic implantation of WT and 81KO MDA-MB-231 cells into NSG mice. Error bars represent standard deviation. *p = 0.011 (*n* = 5 mice). (**H**) Quantification of tumor weights, lung metastases, and relative metastatic burden normalized by tumor weight. Error bars represent standard deviation. Two-tailed Student *T*-test was used. ***p = 1e−07, *p = 0.011. (**I**) Table of mouse mortality or survival by day 32 after orthotopic implantation. Three mice from the WT group died and the left two were sick and sacrificed whereas the 81KO tumor-bearing mice would have survived. Fisher's test was used **p = 0.01 (*n* = 5). (**J**) BLI images of lung colonization following the tail vein injections of MDA-MB-231 WT and 81KO cells into NSG mice on days 0 and 26. The bottom row shows dissected lungs ex vivo. (**K**) Quantified BLI signals (left panel) and normalized metastasis intensity (relative to the day 0 signals) of MDA-MB-231 cells in the dissected lungs ex vivo on day 26 after tail vein injections. Error bars represent standard deviation. Two-tailed Student *T*-test **p < 0.007. ***p = 0.001 (*N* = 3 mice). (**L**) Representative fluorescence images (top two panels) of mouse lungs and quantified metastatic colonies (singles and clustered, bottom panel) of WT and CD81KO L2G$^+$ MDA-MB-231 cells (D26). Scale bar = 100 mm. Error bars represent standard deviation. Two-tailed Student *T*-test was used. *p = 0.044. BLI images of mice (days 0 and 1) and dissected lungs on day 1 (**M**) lood circulating tumor cell (CTC) counts (L2G$^+$ singles and clusters) measured via flow cytometry on day 1 (**N**), and relative lung metastasis via BLI on day 1 (**O**, relative to day 0) following the tail vein injections of 4T1-WT and Cd81KO cells into Babl/c mice (*N* = 3), *p = 0.047, **p = 0.007. Error bars represent standard deviation. Two-tailed Student *T*-test was used. These experiments were repeated at least twice to show consistent conclusions.

The online version of this article includes the following figure supplement(s) for figure 6:

**Figure supplement 1.** CD81 KD inhibits breast cancer cell metastasis.

**Figure supplement 2.** Schematic summary.

associated with lysosome degradation of both CD44 and CD81 in fibroblast cells (*Bartee et al., 2010*) and/or TNBC cells (*Chen et al., 2021*), suggesting CD44 and CD81 might follow similar fates of protein degradation or recycling in both cancer cells and other cells. Nevertheless, both CD44 and CD81 are required for optimal self-renewal and metastasis, emphasizing the indispensable functions of CD44 and CD81 in cell adhesion and intercellular interactions in metastasis. Our phosphoproteome analyses reveal many shared and unshared components between CD44 and CD81 signaling pathways that regulate endocytosis, endosomes, lysosomes, and exocytosis.

Several studies suggest tumor recurrence often occurs due to an increased number of CTCs, some of which display tumor-regenerative plasticity and reprogramming phenotypes (*Giordano et al., 2012*; *Książkiewicz et al., 2012*; *Aktas et al., 2009*), or transform into tumor-initiating cells (*Dahan et al., 2014*; *Dawood et al., 2014*; *Giuliano et al., 2011*; *Pierga et al., 2012*). Our studies demonstrated that cancer exosomes can be part of the transforming factors upon uptake by the CTCs. Here we show that the mammosphere-promoting functions of exosomal CD44 and CD81 illustrate the cross-talk between tumor-initiating cells and surrounding cancer cells that potentially contributes to their self-renewal and/or plasticity. Still, it is necessary to consider that higher concentrations of EV than physiological conditions were used and continues to be a limitation of simplified models frequently used in the field. While CD44 remains understudied in exosome biogenesis, a CD44 variant has been reported to be involved in interluminal vesicle loading (*Wang et al., 2018*) which may be linked to maintaining tumor-initiating cells and tumor progression. Furthermore, a subset of CTCs and tumor-initiating cells may exhibit dynamic changes in epithelial–mesenchymal transition phenotype (*Yu et al., 2013*), which can upregulate CD81 expression in mesenchymal breast cancer (*Uretmen Kagiali et al., 2019*). While the role of CD81 displayed on CTCs is largely understudied, previous studies have used CD81$^+$/CD56$^+$/CD45$^-$ markers to detect neuroblastoma cells in the peripheral blood of patients (*Nagai et al., 2000*). Future studies will be needed to investigate the effects of cancer exosomes on tumor stromal cells and immune cells in various microenvironment niches, not only in TNBC, but also in other cancers as well.

## Materials and methods

### Human specimen analyses

All human blood and tumor specimen analyses complied with NIH guidelines for human subject studies and were approved by the Institutional Review Boards at Northwestern University (STU00203283).

The investigators obtained written informed consent from all subjects whose blood specimens were analyzed.

## Animal studies

All mice used in this study were kept in specific pathogen-free facilities in the Animal Resources Center at Northwestern University. All animal procedures complied with the NIH Guidelines for the Care and Use of Laboratory Animals and were approved by the respective Institutional Animal Care and Use Committees (IACUC: IS00014165). Six- to twelve-week female mice were purchased from Jackson Laboratories (cat# 005557 for NSG mice and cat# 000651 for Balb/c mice), randomized by age and weight. Mice were grouped in a blinded manner and excluded from experiments due to sickness or conditions unrelated to tumors. Sample sizes were determined based on the results of preliminary experiments, and no statistical method was used to predetermine sample size. All PDX tumors were established, and orthotopic tumor implantation was performed as described previously (*Liu et al., 2010*; *Al-Hajj et al., 2003*). For tumorigenic assays, cells and Matrigel (Corning 354234) were implanted into the mammary fat pad of mice at low density (1000–10 cells). Mice were monitored using the Lago in vivo imaging system. For artificial metastasis experiments, (100,000) tumor cells were injected into the mice tail vein and imaged using the Lago in vivo imaging system. For spontaneous metastasis experiments using MDA-MB-231 (ATCC) cells, 10,000 cells were implanted orthotopically in NSG mice. For spontaneous metastasis experiments using 4T1 cells, 1000 WT cells and 6000 Cd81KO cells were implanted orthotopically in Balb/c mice.

## Blood collection and CTC analysis

This study was approved by the Institutional Review Board (IRB) of Northwestern University. All participants signed informed consent forms. About 8–10 ml of whole blood was collected from breast cancer patients into a 10 ml CellSave Preservative tube containing a cellular fixative (Janssen Diagnostics, LLC, Raritan, NJ). Blood specimens were maintained at room temperature and processed within 96 hr of being drawn. CTC analysis was performed using CellSearch CTC kits on the FDA-approved CellSearch System (Item 7900001, Menarini Silicon Biosystems, Inc). The kit uses ferrofluid nanoparticles with antibodies that target epithelial cell adhesion molecules (EpCAM) to magnetically separate CTCs from the bulk of other cells in the blood. CTCs were identified by positive staining for both cytokeratins (CK) and DAPI and negative staining for CD45 (CK$^+$/DAPI$^+$/CD45$^-$). CTC clusters were defined as an aggregation of two or more CTCs containing distinct nuclei and intact cytoplasmic membranes. To determine the expression of CD81 on CTCs, the PE-conjugated anti-CD81 antibody (BD 555676, RRID:AB_396029) was also added. CTCs were also analyzed using flowcytometry by gating single cells and cell clusters for CD45 negative (BD 557748, RRID:AB_396854) and DAPI negative (Sigma D9542). The proportion of CTC events was calculated by dividing the population of interest by the sum of total CTC events. For example, to calculate the percentage of CD44$^+$CD81$^+$CTCs in a patient, CD44$^+$CD81$^+$ events were divided by the sum of CD44$^+$CD81$^+$, CD44$^+$CD81$^-$, CD44$^-$CD81$^+$, and CD44$^-$CD81$^-$ events.

## Cell culture, CRISPR gene knockout, and transfections

MDA-MB-231 and HEK293ft cells were purchased from ATCC, and periodically verified to be *Mycoplasma* negative using MycoAlert Mycoplasma Detection Kit (Lonza cat #LT07-218). Cell morphology, growth characteristics, and microarray gene-expression analyses were compared with published information to ensure their authenticity. Early passages of cells (<20 passages) were maintained in Dulbecco's modified Eagle medium (Sigma-Aldrich D6429) with 10% fetal bovine serum (FBS) (Fisher Scientific SH30071.03) + 1% penicillin–streptomycin (P/S) (Sigma-Aldrich P4333). Pooled populations of WT control cells, CD81KO MDA-MB-231 cells, and Cd81KO 4T1 cells were made using multiple lentiviral gRNAs with a BFP reporter for each gene (Sigma Cat# HS5000016583 and HS5000016584 for human *CD81*, MM5000007469 and MM5000007470 for mouse *Cd81*, and gRNA control vector) and CRISPR-Cas9 with a GFP reporter (Sigma cat# CMV-CAS9-2A-GFP), and flow sorted based on expression of GFP and BFP reporters and absence of CD81/Cd81. CD44KO cells were generated based on the validated lentiviral gRNAs and protocol described previously (*Liu et al., 2019*) and then combined with CD81KO for generation of dKO cells (pooled populations). Stable CD81 overexpression CD81KO MDA-MB-231 cells were made using pDUAL *CD81* GFP lentivirus (addgene #86980;

RRID:Addgene_86980). pDUAL-eGFP (addgene #63215; RRID:Addgene_63215) was used as a negative control.

For exosome experiments, FBS was exosome-depleted by ultracentrifugation at $100,000 \times g$ for 16 hr at 4°C. Primary tumor cells were cultured in HuMEC-ready medium (Thermo Fisher Scientific 12752010) plus 5% FBS and 0.5% P/S in collagen type I (Fisher Scientific – Corning Corning 354231) coated plates. Pooled siRNAs (SMART pool with 3–4 siRNAs; Dharmacon CD81 Cat# L-017257-00, Dharmacon negative control non-targeting pool Cat# D-001810-10-50) were transfected using Dharmafect (Dharmacon) at 100 nmol/l. For overexpression experiments in HEK293ft cells, pCMV6-FLAG-*CD44* (OriGene), pCMV3-HA-*CD81* (Sino Biological HG14244-NY), pCMV3-SP-N-HA (Sino Biological CV021), pDUAL GFP (addgene #86980; RRID:Addgene_86980), pDUAL e-GFP (addgene #63215; RRID:Addgene_63215) plasmids were transfected into cells using Fugene HD (Promega E231A) or jetPRIME (Polyplus 114-01). After 48 hr (or 96 hr, if double transfection), cells were collected for immunoprecipitation and immunoblotting or other assays.

## Isolation and purification of exosomes or small extracellular vesicles from cells

Exosomes or small extracellular vesicles (sEVs) were isolated from the cell culture supernatant as described previously (*Kibria et al., 2016*). Briefly, the cells were cultured as monolayers for 48 hr in complete medium under an atmosphere of 5% $CO_2$ at 37°C. When cells reached a confluency of approximately 80%, exosomes were isolated by differential centrifugation. First, the culture supernatant was centrifuged at $2000 \times g$ for 10 min followed by 30 min centrifugation at $10,000 \times g$ to remove dead cells and cell debris. The supernatant was ultracentrifuged for 70 min at $100,000 \times g$ using an SW28 rotor to pellet the exosomes/sEVs. Exosomes/sEVs were washed by resuspension in 30 ml of sterile PBS (GE Healthcare SH30256.01), and pelleted by ultracentrifugation for 70 min at $100,000 \times g$. The final exosome/EV pellet was resuspended in 100 µl PBS and stored at −80°C.

## Mammosphere formation, cell clustering, and migration assays

For mammosphere assays, cells were seeded into 6- or 12-well plates at a concentration of 2000 or 1000 cells per well in replicates of 3 or 4 using Prime-XV Tumorsphere Serum Free Media (Irvine Scientific 91130). Cells were monitored for up to 17 days, when spheres were imaged and counted to assess mammosphere formation capacity. For exosome or sEV education experiments, cells were seeded into 12-well plates at a concentration of 50,000 cells and treated with 10–15 µg exosomes/sEVs every other day for 1–2 weeks. The cells were split as necessary and mammosphere formation assay followed (2000 cells seeded per well). For clustering assays, cells were seeded into polyhema-coated 96-well plates at a concentration of 25,000 cells per well for cell lines and 100,000 cells per well for primary cells. Cells were monitored up to 72 hr and analyzed by Incucyte Imaging System software. For migration assays, cells were seeded into 96-well Image-Lock plates at 30,000 cells per well. After 16 hr, a scratch wound was introduced into each plate using the Incucyte Wound Maker and then monitored by Incucyte for up to 48 hr to visualize wound closure.

## EV education

The EV education/reprogramming doses were determined based on literature reports and our own measurements. From published reports (*Matsumoto et al., 2016*), high levels of EVs could be detected in human/mouse plasma of the patients/mice with cancer ($>1 \times 10^9$/ml higher in patients or mice with cancer than non-cancer controls). In our studies, TNBC cells do secret large amounts of EVs, with a yield of ~$2 \times 10^8$ EVs/ml supernatant (MDA-MB-231 culture medium) or ~$1 \times 10^9$ EVs /ml (4T1 cell culture) as measured on NTA as well as Apogee MFV. Please note the small EV collecting efficiency is only ~10% by ultracentrifugation at $100,000 \times g$ for 70 min during the EV purification and we end up obtaining 1–2% EVs after two spins (one or two washes) (200 ml culture to yield 60–75 µg EV protein). Therefore, we utilized an amount of the EVs purified from 50 times larger volume of culture supernatants for EV educating culture. For example, 10–15 µg purified EVs purified from ~50 ml supernatant were used for 1 ml educating culture to make $2–5 \times 10^8$ EVs/ml for reprograming CD81KO cells. The dose is pathophysiologically relevant to the EV concentrations in the plasma and culture media.

## Immunofluorescence

Cells were seeded onto 4-chamber slide wells at a concentration of 20,000 cells per well. Clustered cells were attached using a cytospin. Cells were fixed with 4% paraformaldehyde and permeated

with 1% Triton X-100 in PBS for 15 min at room temperature. Cells were then washed three times for 5 min each with 0.05% PBS and blocked with 10% bovine serum albumin (BSA) in PBS for 60 min. Cells were then washed again and primary antibody was added overnight at 4°C (CD81 Millipore HPA007234 [RRID:AB_1846333] 4 µg/ml and CD44 MA513890 [RRID:AB_10986810] 2 µg/ml). After washing, secondary antibody was added for 60 min at room temperature (Texas Red T862 Thermo Fisher (RRID:AB_2556781) and Alexa-488 A11008 Thermo Fisher (RRID:AB_1431650)). Finally, the cells were washed, and a cover slide was placed with mounting media. The slides were imaged using Nikon A1R (A) Spectral.

## Structural modeling

The structure of CD44 (the shortest standard isoform X4) was predicted by the iTasser webserver (*Yang et al., 2015*), with abundant homologous structures available to the N-terminal extracellular domains. And the structure of CD81 in a 'closed' conformation was obtained from the Protein Data Bank (PDB ID: 5TCX), with a missing extracellular loop (residues 38–54) inserted (*Zimmerman et al., 2016*). The two structures were first rigidly docked while being biased toward extracellular regions, using the ClusPro webserver (*Kozakov et al., 2017*). The resulting structural models of CD44–CD81 complexes were then flexibly refined, using the software BAL (*Cao and Shen, 2020*), where binding hotspots (in probabilities from 0 to 1) and binding affinity (in kcal/mol) are predicted and weighted-averaged over all structural models. Two short extracellular helices (resi. 160–170 and 181–187) were predicted to be enriched in binding hotspots and their borders include C156 and C190 forming a disulfide bond. Therefore, a form of CD81 with S159-K187 deletion (CD81d) was suggested to impair its interaction with CD44 while maintaining its stability. The structure of CD81d was again predicted with iTasser and the docking and analyses of CD44/CD81d followed the protocol described above.

## Flow cytometry and cell sorting

To detect cell surface proteins, cells were first blocked with mouse IgG 1 (Cat# l5381, Sigma, St. Louis, MO, USA) for 10 min on ice. Cells were then incubated with antibody (BD Biosciences, San Jose, CA, USA) for 20 min on ice, washed, and analyzed using a BD LSR-2 flow cytometer or BD Aria cell sorter (BD Biosciences).

## RNA sequencing

MDA-MB-231 cells were seeded and transfected with non-targeting siRNA (siCon) and si*CD81* (pooled, Dharmacon) using Dharmafect. After 48 hr, the cells were harvested and submitted for RNA sequencing. Total RNA of MDA-MB-231 cells was isolated using Trizol, phase separated by chloroform, and extracted by alcohol. Samples were sent to Northwestern University's Center for Genetic Medicine Sequencing core facility for deep sequencing analysis. RNA sequencing was performed on a HiSeq 4000, and a library was made using aTruSeq Total RNA-Seq Library Prepkit. Data were processed and quantified using STAR (*Dobin et al., 2013*), DESeq2 (*Love et al., 2014*), and HTSeq (*Anders et al., 2015*). Analysis of differentially expressed genes was set to a cutoff of false discovery rate (FDR) <0.05 and log2 (fold change) >0.48 or <−0.48. Finally, the pathway analysis of significantly differentially expressed genes was obtained by using Metascape (http://metascape.org) (*Tripathi et al., 2015*). The raw data files of RNA-seq data generated with control and si*CD81*-transfected MDA-MB-231 cells have been deposited to GEO database with accession number GSE174087.

## Mass spectrometry of tumor cells and exosomes/sEVs

Exosomes or sEVs were isolated from MDA-MB-231 WT control and CD44KO cell cultures via standard ultracentrifugation as described (*Kibria et al., 2016*). The cells and exosomes/EVs were lysed with 2% sodium dodecyl sulfate and protease inhibitor cocktail. Proteins were extracted using pulse sonication, and cleaned up by filter-aided sample preparation to remove detergents. After LysC/Trypsin digestion, 500 ng proteins were analyzed via 4 hr LC/MS/MS method at Case Western Reserve University Proteomics Core facility and the data processed using MetaCore. The fold change was calculated based on total unique spectrum counts.

# Global and phosphoproteome analyses of tumor cells transfected with si*CD81* and si*CD44*

For protein extraction, cell pellets were resuspended in cell lysis buffer (100 mM $NH_4HCO_3$, pH 8.0, 8 M urea, 1% protease and phosphatase inhibitor, pH 8.0) and protein concentrations were measured with a Pierce BCA protein assay (Thermo Fisher Scientific). Proteins were reduced with 5 mM dithiothreitol for 1 hr at 37°C and alkylated with 10 mM iodoacetamide for 45 min at 25°C in the dark. Protein was digested with Lys-C (Wako) (1:50 enzyme-to-substrate ratio) for 3 hr at 25°C and with sequencing-grade modified trypsin (Promega, V5117) at 25°C for 14 hr. After digestion, each sample was desalted by C18 SPE extraction and concentrated for BCA assay to evaluate the peptide yield.

The tryptic peptides from bulk samples were dissolved with 50 mM HEPES (4-(2-hydroxyethyl)-1-p iperazineethanesulfonic acid) (pH 8.5) and then mixed with a TMTpro reagent in 100% ACN (Aceto-nitrile). A ratio of TMTpro to peptide amount of 4:1 was used. After incubation for 1 hr at room temperature, the reaction was terminated by adding 5% hydroxylamine for 15 min. The TMTpro-labeled peptides were then acidified with 0.5% FA. Peptides labeled by different TMTpro reagents were then mixed, dried using Speed-Vac, reconstituted with 3% acetonitrile, 0.1% formic acid, and desalted on C18 SepPak SPE columns.

Peptide fractionation by bRPLC and phosphopeptides enrichment by IMAC were performed as previously reported (*Mertins et al., 2018*). Lyophilized global and phosphorylated peptides were reconstituted in 12 µl of 0.1% FA (Formic acid) with 2% ACN and 5 µl of the resulting sample was analyzed by LC–MS/MS using an Orbitrap Fusion Lumos Tribrid Mass Spectrometer (Thermo Scien-tific) connected to a nanoACQUITY UPLC system (Waters Corp., Milford, MA) (buffer A: 0.1% FA with 3% ACN and buffer B: 0.1% FA in 90% ACN) as previously described (*Tsai et al., 2020*). Peptides were separated by a gradient mixture with an analytical column (75 µm i.d. × 20 cm) packed using 1.9 µm ReproSil C18 and with a column heater set at 50°C. Peptides were separated by a gradient mixture: 2–6% buffer B in 1 min, 6–30% buffer B in 84 min, 30–60% buffer B in 9 min, 60–90% buffer B in 1 min, and finally 90% buffer B for 5 min at 200 nl/min. Data were acquired in a data-dependent mode with a full MS scan (*m/z* 350–1800) at a resolution of 60 K with AGC setting set to $4 \times 10^5$ and maximum ion injection period set to 50 ms. The isolation window for MS/MS was set at 0.7 *m/z* and optimal HCD fragmentation was performed at a normalized collision energy of 30% with AGC set as $1 \times 10^5$ and a maximum ion injection time of 105ms. The MS/MS spectra were acquired at a resolution of 50 K. The dynamic exclusion time was set at 45 s. Raw datasets have been deposited in the Japan ProteOmeSTandard Repository (https://repository.jpostdb.org/) (*Okuda et al., 2017*). The accession numbers are PXD029529 for ProteomeXchange (*Vizcaíno et al., 2014*) and JPST001321 for jPOST. The access link is https://repository.jpostdb.org/preview/1370203119618182ba1c0f2 with access key 7811 for reviewer only until accepted.

The raw MS/MS data were processed with MSFragger via Fragpipe (*Kong et al., 2017*; *Teo et al., 2021*) with TMT16 quantitation workflow. The MS/MS spectra were searched against a human UniProt database (fasta file dated July 31, 2021 with 40,840 sequences which contain 20,420 decoys). The intensities of all 16 TMT reporter ions were extracted from Fragpipe outputs and analyzed by Perseus (*Tyanova et al., 2016*) for statistical analyses. The abundances of TMTpro were firstly log2 transformed. The TMT intensities were normalized based on the column-centering by median values for statistical pairwise comparison. For pathway analysis, the significantly expressed protein or phosphophoproteins after ANOVA *T*-test analysis (FDR <0.01) were analyzed by DAVID (*Jiao et al., 2012*). The protein–protein interaction analysis was performed with STRING (*Szklarczyk et al., 2021*) and Cytoscape (*Shannon et al., 2003*).

MaxQuant was used to process the raw MS/MS data with 20,198 sequences recognized against a human UniProt database (fasta file dated April 12, 2017), default setting for mass tolerance for precursor and fragment ions and 'Reporter ion MS2' for isobaric label measurements. A peptide search was performed with Trypsin/P and allowed a maximum of two missed cleavages. Carbami-domethyl (C) was set as a fixed modification; acetylation (protein N-term), oxidation (M) and Phospho (STY) were set as variable modifications for phosphoproteome analysis. The FDR was set to 1% at the level of proteins, peptides, and modifications. The Phospho (STY) Sites.txt file was used for further quantitation. The intensities of all 10 TMT reporter ions were extracted from MaxQuant outputs and analyzed by Perseus (*Tyanova et al., 2016*) for statistical analyses.

## Transmission electron microscopy

Cells were harvested from the medium and washed with Na/K Phosphate buffer (0.1 M, pH 7.2). Primary fixation was done with 3% glutaraldehyde for 2 hr at 4°C. The cells were then post-fixed in 1% osmium tetroxide for 1 hr at 4°C. The cells were resuspended in molten agar (2%). Small blocks of solidified agar (1sq.mm) were cut and passed through series of 30%, 50%, and 90% ethanol (vol/vol) for 15 min each. The cells were further dehydrated with 100% ethanol (30 min × 3). The dehydrated agar blocks were suspended in propylene oxide for 20 min at room temperature and then treated with 1:1 mixture of propylene oxide and Epon-812 for 1 hr at room temperature and Epon-812 for 4 hr at room temperature. Blocks were embedded with Epon-812 for 48 hr at 60°C. Ultra-thin sections were cut with a Leica UC6 ultramicrotome and examined with a FEI Tecnai Spirit transmission electron microscope (FEI, Hilsboro City, OR, USA).

## Cryo-electron microscopy

For cryoEM visualization, samples were prepared from freshly isolated exosomes/sEVs at 0.25 µg/µl concentration. For cryo-freezing, 3.5 µl of exosome solutions were applied to fresh glow-discharged (10 s, 15 mA; Pelco EasiGlow) lacey carbon TEM grids (Electron Microscopy Services) and vitrified using a FEI Vitrobot Mark IV (FEI, Hillsboro, OR). The sample was applied to the grid and kept at 85% humidity and 10°C. After a 10-s incubation period the grid was blotted with Whatman 595 filter paper for 3.5 s using a blot force of 5 and plunge frozen into liquid ethane. Samples were imaged using a JEOL 3200FS electron microscope equipped with an omega energy filter operated at 300 kV with a K3 direct electron detector (Ametek) using the minimal dose system. The total dose for each movie was ~10 e$^-$/A2 at a nominal magnification between 8000 (pixel size 4.1 Å).

## NanoSight particle tracking and rapid microflow cytometer analysis of exosomes or EVs

The size and particle count of exosomes/EVs were measured using NanoSight NS3000, a nanoparticle tracking analyzer (NanoSight Ltd, Malvern, United Kingdom). Exosomes/EVs (5 µg) were diluted in 1 ml PBS and then processed. Similarly, direct supernatants after removal of cell debris or purified exosomes/EVs diluted in 300 µl of PBS were loaded to the Apogee micro flow vesiclometer for EV count analysis and normalized based on the cell numbers to compare the EV secretion efficiency or yield.

## Immunoblot analysis

Cells and exosomes/EVs were lysed using RIPA lysis buffer (cell signaling 9806S). Protein-containing lysates of exosomes/EVs (5 µg) were run on a 4–20% Mini-PROTEIN TGX gel (Bio-Rad 4561096) and transferred to a nitrocellulose membrane. The blots were incubated separately either with mouse monoclonal anti-human CD44 (156-3 C11) antibody (Cat# MA513890, Thermo Fisher Scientific, Waltham, MA, USA, RRID:AB_10986810), mouse monoclonal anti-human CD63 antibody (Cat# ab8219, Abcam, Cambridge, MA, USA, RRID:AB_306364) at a dilution of 1:1000, Rabbit polyclonal anti-human CD81 (Cat# GTX101766, Genetex, Irvine, CA, USA, RRID:AB_10618892) at a dilution of 1:1000, rabbit polyclonal anti-human Grp94 (Cat# 2104 P, Cell Signaling Technology, Danvers, MA, USA) at a dilution of 1:1000, or mouse monoclonal anti-human β-actin (Cat# ab8224, Abcam, Cambridge, MA, USA; RRID: AB_449644) at a dilution of 1:1000 (in Tris-buffered saline [TBS] containing 2% BSA) at room temperature for 1 hr, followed by washing with TBS buffer. The blots were incubated with secondary antibody (horseradish peroxidase-conjugated goat anti-mouse [Cat# W402B] or goat anti-rabbit IgG [W401B] from Promega, Madison, WI, USA) at a dilution of 1:10,000 (in 2% BSA containing TBS) for 1 hr at room temperature. The blots were treated with an enhanced chemiluminescence kit according to the user manual and developed using a ChemiDoc MP Imaging System (Bio-Rad).

## Co-immunoprecipitation

For endogenous co-immunoprecipitation, cells were lysed and preincubated with Protein A/B PLUS agarose beads (Cat# sc-2003, Santa Cruz Biotech, Dallas, TX, USA) for 2 hr. The supernatant was removed, and the protein concentration was measured. Then 100 µg of cell lysate was incubated with CD44 anti-body or bead control overnight and then Protein A/B PLUS agarose beads overnight. The beads were washed five times and denatured with 4× Laemmli sample buffer (Cat: 161-0747, Bio-Rad,

Hercules, CA, USA) at 100°C for 5 min. For exogenous co-immunoprecipitation, cells were lysed, and protein concentration was measured. Then 200 µg of cell lysate was incubated with Anti-FLAG M2 Magnetic Beads, Sigma-Aldrich (Cat# M8823, Sigma-Aldrich, St. Louis, MO, USA) overnight. The next day the beads were washed, and the beads were eluted using glycine. The elution was then combined with 4× Laemmli sample buffer and denatured at 100°C for 5 min.

## Immunohistochemistry

Mouse xenograft lung tissues or patient primary tumors were paraffin-embedded and sectioned by routine techniques. Heat-induced antigen retrieval was achieved using Decloaker solution for 15–20 min (Biocare Medical, RD913L). Tissue sections were blocked with TBS/10% NGS, then incubated with CD81 (Cat # HPA007234 Millipore Sigma, St. Louis, MO, USA; RRID:AB_1846333) or CD44 (Cat# MA513890, Thermo Fisher Scientific, Waltham, MA, USA; RRID:AB_10986810) primary antibody overnight, followed by Dako envision plus kit and DAB staining (Dako K4010). All samples were counterstained with hematoxylin.

## RNA extraction and real-time PCR

Total RNA was extracted using Trizol (Invitrogen), and RNA was paecipitated with isopropanol and glycogen (Invitrogen). After reverse transcription reactions, real-time PCR for genes was performed using individual gene Taqman primers (Applied Biosystems) with an ABI 7500 real-time PCR system. GAPDH was used as a control.

## Kaplan–Meier plots

Kaplan–Meier overall survival, relapse free survival, and distant metastasis free survival plots for protein expression of CD81 were made using Kaplan_Meier Plotter (kmplot.com) (*Lánczky and Győrffy, 2021*). The dataset Liu_2014 (*n* = 126) was used. All patients had TNBC.

## Breast tumor TMA

A total of 89 formalin-fixed paraffin-embedded breast tumor tissues were included on the tumor TMA with selected tumor regions guided by hematoxylin–eosin staining images. The demographic and clinical characteristics of these tumors are included in *Figure 4—figure supplement 2*. To make the TMA that allows microscopic comparison of the staining characteristics of different blocks and prevents exhaustion of pathological material, a core of paraffin was removed from a 'recipient' paraffin block (one embedded without tissue) and the remaining empty space is filled with a core of paraffin-embedded tissue from a 'donor' block. A donor block H&E that is representative of the tissue remaining in the block was used to select the sample core with a color marker corresponding to tumor, benign, etc. Matched blocks were pulled out and a recipient TMA block was made and trimmed well with the face of the block even with a size of 1.5 mm core by using the semi-automatic Veridiam Tissue Microarryer VTA-100. The created TMA block was sectioned for staining. In this TMA, 19 cases from NU 16B06, 9 ER negative cases, 30 triple negative cases, 27 ER-positive cases, and 4 normal breast cases were selected and constructed on the recipient block.

## Statistical analysis

For all assays and analyses in vitro, unless otherwise specified, a two-tailed Student's *T*-test performed using Microsoft Excel was used to evaluate the p values, and p < 0.05 was considered statistically significant.

## Material sharing and data availability

All data generated or analyzed during this study are included in the manuscript and supporting files. Newly created material and cell lines, such as human CD81KO MDA-MB-231 cells and mouse Cd81KO 4T1 cells are available upon request. The RNAseq and proteomic datasets have been deposited as specified above. Source Data Files of uncropped full western blots have been provided for all blots.

## Acknowledgements

We are grateful for the tremendous support by Northwestern University Core facilities, including but not limited to the CTC Core, the Center for Comparative Medicine, Flow Cytometry Core, Small Animal Imaging, Microscopy Imaging, NUSeq, Bioinformatics, Mouse Histology & Phenotyping Laboratory, and Pathology Core. We also thank Case Western Reserve University Mass Spectrometry Core and Cancer Center Small Animal Facilities for their support. This project has been partially supported by the Department of Defense grant W81XWH-16-1-0021 and W81XWH-20-1-0679 (H Liu), the NIH/NCI grants R01CA245699 (H Liu and EK Ramos); NIH/NIGMS R35GM124952 (Y Shen) and R01GM139858 (T. Shi); National Science Foundation CCF-1943008 (Y Shen); the Lynn Sage Cancer Research Foundation (X Liu, M Cristofanilli, and H Liu), Susan G Komen Foundation CCR18548501 (X Liu); American Cancer Society ACS127951-RSG-15-025-01-CSM (H Liu); Northwestern University start-up grant (H Liu), and NIH Fellowships T32 CA009560 (EK Ramos), T32 CA080621-15 and the Julius Kahn Fellowship (R Taftaf), and T32GM008061 (EJ Schuster). Portions of this research were conducted with the advanced computing resources provided by Texas A&M High Performance Research Computing.

## Additional information

### Competing interests

Erika K Ramos, Nurmaa K Dashzeveg: has patents on exosomes which are not related to this manuscript. Andrew D Hoffmann, Lamiaa El-Shennawy, Emma J Schuster: has patents on exosomes which are not related to the paper. Huiping Liu: is scientific co-founder of ExoMira Medicine, Inc and has patents related to exosome therapeutics which are not related to the scientific discoveries of this paper. The other authors declare that no competing interests exist.

### Funding

| Funder | Grant reference number | Author |
| --- | --- | --- |
| National Cancer Institute | R01CA245699 | Erika K Ramos Huiping Liu |
| National Institute of General Medical Sciences | R35GM124952 | Yang Shen |
| U.S. Department of Defense | W81XWH-16-1-0021 | Huiping Liu |
| Susan G. Komen | CCR18548501 | Xia Liu |
| American Cancer Society | ACS127951-RSG-15-025-01-CSM | Huiping Liu |
| National Cancer Institute | T32 CA009560 | Erika K Ramos |
| National Cancer Institute | T32GM008061 | Emma J Schuster |
| National Science Foundation | CCF-1943008 | Yang Shen |
| National Institute of General Medical Sciences | R01GM139858 | Tujin Shi |

The funders had no role in study design, data collection, and interpretation, or the decision to submit the work for publication.

### Author contributions

Erika K Ramos, Data curation, Formal analysis, Validation, Investigation, Visualization, Methodology, Project administration; Chia-Feng Tsai, Resources, Data curation, Validation, Investigation, Methodology, Writing – original draft, Project administration, Writing – review and editing; Yuzhi Jia, Resources, Data curation, Formal analysis, Validation, Investigation, Methodology; Yue Cao, Resources, Software, Investigation, Methodology; Megan Manu, Data curation, Formal analysis, Validation, Investigation, Methodology; Rokana Taftaf, Valery Adorno-Cruz, Emma J Schuster, David Scholten, Dhwani Patel,

Xia Liu, Data curation, Validation, Investigation, Methodology; Andrew D Hoffmann, Data curation, Software, Validation, Investigation, Methodology; Lamiaa El-Shennawy, Resources, Data curation, Validation, Investigation, Methodology; Marina A Gritsenko, Validation, Investigation, Methodology; Priyam Patel, Software, Validation, Investigation; Brian Wray, Resources, Software, Validation, Investigation, Methodology; Youbin Zhang, Resources, Data curation, Investigation, Methodology; Shanshan Zhang, Data curation, Investigation, Visualization, Methodology; Ronald J Moore, Jeremy V Mathews, Data curation, Investigation, Methodology; Matthew J Schipma, Resources, Software, Methodology; Tao Liu, Resources, Data curation, Supervision; Valerie L Tokars, Resources, Data curation, Investigation, Visualization, Methodology; Massimo Cristofanilli, Resources, Supervision, Funding acquisition, Investigation; Tujin Shi, Resources, Data curation, Supervision, Investigation, Methodology, Writing – review and editing; Yang Shen, Conceptualization, Resources, Data curation, Software, Supervision, Investigation, Methodology, Writing – original draft, Writing – review and editing; Nurmaa K Dashzeveg, Conceptualization, Data curation, Supervision, Investigation, Visualization, Methodology, Writing – review and editing; Huiping Liu, Conceptualization, Resources, Formal analysis, Supervision, Funding acquisition, Investigation, Project administration, Writing – review and editing

## Author ORCIDs
Erika K Ramos http://orcid.org/0000-0003-3194-3442
Chia-Feng Tsai http://orcid.org/0000-0002-6514-6911
Andrew D Hoffmann http://orcid.org/0000-0002-5479-944X
Marina A Gritsenko http://orcid.org/0000-0001-9992-9829
Dhwani Patel http://orcid.org/0000-0002-9368-2850
Priyam Patel http://orcid.org/0000-0002-9433-5017
Ronald J Moore http://orcid.org/0000-0003-2806-2855
Matthew J Schipma http://orcid.org/0000-0002-0865-1057
Tao Liu http://orcid.org/0000-0001-9529-6550
Valerie L Tokars http://orcid.org/0000-0002-2619-3969
Yang Shen http://orcid.org/0000-0002-1703-7796
Nurmaa K Dashzeveg http://orcid.org/0000-0002-6702-5187
Huiping Liu http://orcid.org/0000-0003-4822-7995

## Ethics
All mice used in this study were kept in specific pathogen-free facilities in the Animal Resources Center at Northwestern University. All animal procedures complied with the NIH Guidelines for the Care and Use of Laboratory Animals and were approved by the respective Institutional Animal Care and Use Committees.

## Decision letter and Author response
Decision letter https://doi.org/10.7554/eLife.82669.sa1
Author response https://doi.org/10.7554/eLife.82669.sa2

---

# Additional files

## Supplementary files
- Supplementary file 1. Overlapping list of 38 proteins in CD44 and siCD44.
- Supplementary file 2. Down or up-regulated genes without cutoffs.
- Supplementary file 3. RNAseq data: down- and up-regulated pathways with 1.5 fold change.
- Supplementary file 4. Global and phosphoproteomics data in cluster versus adherent cells.
- Supplementary file 5. KEGG Pathways of global proteome clusters (G1 to G6) and phosphoproteome cluster (P1 to P6).
- Supplementary file 6. Heatmap for siCD81 and siCD44 global proteomics in clusters.
- Supplementary file 7. Phosphoproteomics data for heatmap in siCD81 and siCD44.
- Supplementary file 8. Differentially Expressed Proteins in MDA-MB-231 WT versus CD44KO cell by mass spectrometry.
- Supplementary file 9. Altered proteins in EVs of MDA-MB-231 WT versus CD44KO by mass spectrometry.

## Data availability

RNA sequencing data have been deposited to GEO database with accession number GSE174087. Mass spec raw datasets have been deposited in the Japan ProteOmeSTandard Repository (JPST001321) and ProteomeXchange (PXD029529).

The following datasets were generated:

| Author(s) | Year | Dataset title | Dataset URL | Database and Identifier |
|---|---|---|---|---|
| Ramos EK, Patel P | 2022 | RNAseq of MDA-MB-231 siCon and siCD81 cells | http://www.ncbi.nlm.nih.gov/geo/query/acc.cgi?acc=GSE174087 | NCBI Gene Expression Omnibus, GSE174087 |
| Liu H, Tsai CF | 2022 | CD81 facilitates tumor cell clustering and metastasis in triple negative breast cancer | http://proteomecentral.proteomexchange.org/cgi/GetDataset?ID=PXD029529 | ProteomeXchange, PXD029529 |
| Liu H, Tsai CF | 2022 | CD81 facilitates tumor cell clustering and metastasis in triple negative breast cancer | https://repository.jpostdb.org/entry/JPST001321 | jPOST, JPST001321 |

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
