## [Editor Report]

This research innovatively depicted that CD81 could form interaction with CD44 and contributes to the self-renewal, and metastatic capacity of breast cancer stem cells. Through large amounts of validation, either through computational or experimental methods, the conclusions are firmly held back that CD81-CD44 interaction is indispensable for circulating tumor cell dissemination and metastasis. The findings have broadened the knowledge of the molecular regulatory network of breast cancer cell stemness and may be significant for scholars who are experts in cancer biology.

---

## [Decision Letter]

[Editors' note: this paper was reviewed by Review Commons.]

---

## [Author Response]

General Statements

We are grateful to the editors and reviewers for providing insightful comments and revision instructions. The authors on the paper have fully addressed all the questions to the best of our capabilities during the pandemic when it is challenging to everyone. As the first author Erika Ramos has just completed the full revision right before she left the laboratory for her new career, we appreciate your consideration and support. Please note that the updated data includes Figure 1-6 (55 panels) and figure supplements (62 panels), in a total of 117 panels, plus 9 Supplementary files.

Point-by-point description of the revisions

We have inserted a point-by-point reply in blue describing the revisions included in the transferred manuscript.

Reviewer #1 Evidence, reproducibility and clarity (Required):In this manuscript, Dr. Huiping Liu and colleagues investigate the role of CD81 in breast cancer metastasis, cancer stem cells, and extracellular vesicles (EVs). CD81 is a tetraspanin protein that has unclear roles in cancer. The authors discover that CD81 can form a complex with CD44 on cell surface and instigate cell clustering (important for CTC dissemination), self renewal and metastasis using multiple cell lines and xenografts. Multi-omic studies led them to CD81-regualted EVs, which they found to play critical roles in driving stemness and cell clustering. Furthermore, they found that CD81 is co-expressed with CD44 and affects patient outcomes. Overall, the novelty of the work is high, and the amount of data is impressive and of high quality. The experiments presented are well controlled and rigorous. However, there are some concerns as listed below:

We appreciate the positive comments on the high novelty and high quality of the work. The concerns below have been addressed with point-to-point answers.

Major:The conclusion of EVs made by TICs to reprogram non-TICs into TICs is based on adding large amounts of EVs isolated from WT tumor cells. The amount may not be achievable in physiological conditions. Further, if the EVs released by TICs are sufficient to reprogram neighboring tumor cells, this event would be expected to self-propagate and all the tumor cells would become TICs.

We are thankful to the reviewer for the valuable thoughts and input. While the EVs used to educate and reprogram non-TICs may seem to be large amounts, the doses are justified as below and in the Method section to address pathophysiologically relevant questions using testable models.

First, we add this section of “Exosome or sEV education” to the Method section.

“The EV education/reprogramming doses were determined based on literature reports and our own measurements. From published reports (reference PMID: 27599779), EVs could be detected in human/mouse plasma with high levels in the patients with cancer (>1x10^9^/mL higher in cancer patients than healthy controls), as well as mouse plasma (>1x10^9^/ml higher EV counts in the tumor bearing mice than non-tumor mice). In our studies, triple negative breast cancer (TNBC) cells do secret large amounts of EVs, with a yield of 0.2~1x10^9^ EVs /mL supernatant (MDA-MB-231 or 4T1 cell culture) as measured on NTA as well as Apogee micro flow vesiclometer (MFV). Please note the small EV collecting efficiency is only ~10% by ultracentrifugation at 100,000xg for 70 min during the EV purification and we end up obtaining 1-2% EVs after two spins (one or two washes) (200 mL culture to yield 60-75 µg EV protein). Therefore, we utilized an amount of the EVs purified from 50x larger volume of culture supernatants for EV educating culture. For example, 10-15 µg purified EVs purified from ~50mL supernatant were used for 1 mL educating culture to make 2~5x10^8^ EVs/mL for reprograming CD81KO cells. The dose is pathophysiologically relevant to the EV concentrations in the plasma and culture media.”

Second, we admit that the boundary and heterogeneity between TICs and non-TICs are more drastic in patient/mouse tumors in vivo than cultured tumor cells in vitro. Considering the complexity that the educational effects of cancer EVs on surrounding cells in vivo interplay with many other variables such as spatial restrictions, EV-dilution by interstitial fluids, EV release into blood, and mixed tumor-suppressing factors in the microenvironment, we have adopted the simplified EV education models in vitro to quantify the reprogramming effects of TIC EVs on nonTIC tumor cells (CD81KO cells). This model is mainly utilized to examine the effects of EV proteins CD44 and CD81 in measurable reprograming activities. Notably, we observed that CD44+CD81+ TICs tend to gradually dominate in culture whereas non-TIC populations gradually lose and thus a high percentage of cells may become tumorigenic TICs. Our data demonstrate that EVeducation effects can contribute to the outcomes in a CD44- and CD81-dependent manner.

The conclusion that CD44 switched from membrane to intracellular locations adherent vs. suspension cultures is not well supported by the data presented in Figure 1E. It appears that the cells expressing membrane CD44 did not change much, but the cells population expressing intracellular CD44 expanded in the suspension culture.

We appreciate the comment and have clarified the observation and the conclusion accordingly in updated Figure 1C-D:

“CD44 was observed mostly on the cytoplasmic membrane in WT MDA-MB-231 cells, both adherent and in suspension, but the intracellular CD44 accumulated specifically in the cells in suspension (P=0.03) (Figure 1C-D).”

Figure 6E-H experiments are a bit flawed. Metastasis degree is unlikely to be proportional to the primary tumor size, so it is not proper to normalize the metastatic burden to the primary tumor weight. These data can be removed without significantly weakening the major conclusions.

Thanks for the suggestion. We have removed the normalized panel of original Figure 6H and updated with tumor weight and lung metastasis signals.

Figure 3J appears to have different lanes pasted together. ¬¬Therefore, it is unclear whether the results could be compared, calling into the question of the statements involving the proteins studied including OCT4, pSTAT3 and FAK.

Sorry for the confusion. We must clarify that all the lanes are from the same blot which has been stripped and reblotted for multiple proteins. The full blots of previous Figure 3J (new Figure 3-figure supplement 3A) are shown in Author response image 1.

**Author response image 1. sa2fig1:** Full blots of Suppl. Figure 8A. Full Immunoblot analyses for OCT4, STAT3, phosphoSTAT3 (pSTAT3), FAK, pFAK, CD44 and β Actin using CD81KO MDA-MB-231 cells educated with PBS or EVs derived from WT, CD81KO, and CD44KO, and CD81KO cells.

The authors state that EVs were comparable between WT and 44ko and 84ko cells partly based on the western blot presented in Figure 3E. While TSG101 appears similar, the other markers appear quite different. This needs explanation.

We conducted three independent immunoblots for EV proteins in Figure 3E and quantified the protein intensities in 3F which demonstrates that both CD44 and CD81 are significantly lost in the EVs of CD81KO cells and CD44KO cells, CD63 slightly reduced in the EVs of CD44KO cells, whereas no significant differences in EV markers TSG101 and LAMP2b comparing the KO EVs with the WT EVs (the levels normalized by β-actin loading control). These data suggest that CD81 and CD44 are required for each other’s localization or packaging to the EV.

The rationale for different cut-offs in Figure 4A-C is unclear. Auto cut-offs should not be used to maximize the detection of difference. The exact public dataset used should be stated.

Thanks for the insightful suggestions. In Figure 4-figure supplement 1, we have added new KaplanMeier plots based on Liu_2014 cohort (TNBC) with cutoffs at lower 25%, median and upper 25% of CD81 protein levels, respectively, most of which show significant differences in overall survival (OS), relapse-free survival (RFS), and distant metastasis-free survival (DMSF) between CD81 high and low groups (CD81 as an unfavorable marker). These are consistent with and strengthening the data of the KM plots with auto cut offs in Figure 4A-C.

MinorThe description regarding the screen leading to CD81 is confusing. Including a diagram may help.

Thanks again and we have added a schematic to Figure 1- figure supplement 1A with the experimental procedure details of sorting CD44 +/- cells and performing CD44 knockdown in PDX tumors for mass spectrometry comparisons. CD81 is one of 38 overlapped proteins differently shown in two comparisons.

Figure 1E, the phrase "ratio of cells expressing" is bit confusing. Did the authors mean % of cells expressing their indicated markers?

Yes, the y axis label is equivalent to % (1=100%) and now specified as “Proportion of CD44+ or CD88+ cells” in Figure 1E.

Cryo-EM shows detects impaired membrane integrity of EVs of 81ko but 44ko. Yet, Figure 2 shows that ko of either 44 or 81 disruptions the localization and therefore the function of the other. Explanation is needed for why 44ko did not affect 81 regulation of EVs.

We appreciate the thoughtful comments. After repeating and reanalyzing the experimental data from multiple cryo-EM and immunoblotting experiments, we updated Figure 2 which shows both CD81KO and CD44KO impaired the membrane integrity of EVs. Furthermore, the immunoblot validated that CD44 is deficient in the Evs of CD81KO cells, suggesting a possible role of CD81 and in recruiting CD44 to EV and strengthening the EV membrane integrity. In the meantime, CD81 localization on the membrane is dependent on the presence of membrane CD44 in Figure 1E.

Figure S8C is not a robust correlation analysis. A scatter plot should be presented.

A scatter plot is included (new Figure 4- figure supplement 2C) with R = 0.29 and p = 0.015.

There are a few typos. For example "cytoplastic membranes" should be cytoplasmic membranes. The following sentence is awkward: "Among the siCD81-upregulated genes, SEMA7a, a glycosylphosphatidylinositol membrane anchor promoting osteoclast and blood cell differentiation (58, 59), when further depleted in CD81KO cells, siSEMA7a partially rescued or restored mammosphere formation in these cells (Supplementary Figure S2D-F),.."

Typos were corrected. The sentence has been updated to:

“Among the siCD81-upregulated genes, SEMA7a, a glycosylphosphatidylinositol membrane anchor promoting osteoclast and blood cell differentiation (50, 51), was depleted in CD81KO cells by siSEMA7a which transfection partially rescued or restored mammosphere formation in these cells (Figure 2-figure supplement 1D-F), suggesting a novel role of SEMA7a in inhibiting self-renewal of breast cancer cells.”

Reviewer #2The experiments are generally well performed and convincing, except that all mouse experiments seem to have been performed only once, with small groups of mice (between 4 and 6 with several sites of tumor injection per mouse). Reproducing at least once these experiments should be shown.

We appreciate the positive comments as well as instructive suggestions. We would like to clarify that the mouse studies in Figures 5 and 6 were originally completed with two to three different models using both human and mouse TNBC cells. While one of the tumorigenic experiments in Figure 5 (where three different models were used) was done once due to the pandemic disruption and increased costs for NSG mice, Figure 6 data was originally generated by multiple in vivo experiments. Nevertheless, we have managed to repeat the in vivo tumorigenic experiments. We updated Figure 5 with new results from 2-3 experiments for each panel/model with increased number of injections to 8-20 in 2-5 mice per group in each of three different tumor models. Furthermore, we also selectively repeated 4T1 experiments in Figure 6M with updated legend showing the experiment was consistently repeated.

“These experiments were repeated at least twice to show consistent conclusions”.

**Author response image 2. sa2fig2:** Repeated Figure 6M. BLI images of dissected lungs on day 1 following the tail vein injections of 4T1-WT and Cd81KO cells into Babl/c mice (n=4) * P = 0.03. Two-tailed student T-test was used.

Another unsatisfying aspect is the claim that CD44 and CD81 are specifically required for secretion of the subtype of EVs called exosomes, which forms in intracellular multivesicular endosomes. This claim is based on the (wrong) assertion that CD81 (together with CD9, CD63 and TSG101) is an exosome marker, based on ref 46,47 published in 2006 and 2011: the field has evolved a lot since then, and it is now becoming clear that CD63 (which normally accumulates in multivesicular endosomes) may be enriched in exosomes, but that neither CD9 nor CD81 are, since they mainly localize at the plasma membrane, and thus probably are released more prominently in small microvesicles (see Kowal, J., et al. (2016). Proc Natl Acad Sci U S A 113: E968. and Mathieu, M., et al. (2021). Nat Commun 12(1): 4389.).

Thanks for the comment on the evolving literature of CD81. To avoid confusion and follow the EV nomenclature guidelines, we used the more acceptable and general term “extracellular vesicles (EVs)” instead of “exosomes” in our manuscript. We have also cited the two new publications in PNAS 2016 and Nat Commu 2021 (ref 38 and 39) with modified statement on CD81 as below.

“We performed mass spectrometry proteomic profiling of TNBC patient-derived xenografts (PDX) tumor cells that cluster and discovered that one of the altered proteins upon CD44 depletion was CD81, a tetraspanin protein enriched in extracellular vesicles (EVs) (38-39).”

In figure 3, the authors show empty internal compartments in cells with deleted CD44 or CD81, which (if these compartments are altered MVBs) should then lead to fewer exosomes recovered extracellularly. However, the authors observe instead more particles recovered from these cells. When analyzing the composition of these EVs, the authors show maybe a decrease in CD63 in EVs from both CD44 and CD81 ko cells, but contradictory effects on the presence of Lamp2 (which is a more convincing marker of late endosome/lysosome-derived exosomes): increased in CD44 but decreased in CD81 ko. These results, however, are not really interpretable, since the authors show only a single Western Blot, thus no evidence of reliable changes in protein composition. In any case, I would suggest that the authors do not, in the current state of the article, try to claim any specific subcellular origin of the EVs affected by CD44 and CD81.

We appreciate the comments and would like to clarify our observations. When the CD44/CD81 KO cells show increased empty internal compartments, it is a sign for loss of cellular mass. Therefore, it might be surprising but also reasonable to link such phenotype with an increased quantity of draining EVs in low quality (disrupted membrane) from CD44KO and CD81KO cells that show active endocytosis and/or exocytosis pathways.

To better quantify the proteins of the EVs derived from WT, CD44KO and CD81KO cells, we repeated the western blots three times and in new Figure 3F we reported the levels of CD44, CD81, and other EV proteins (CD63, TSP101, and LAMP2b). Compared to WT EVs, CD44 and CD81 are relatively compromised in the EVs of CD81KO and CD44KO, respectively. CD63 slightly decreases in CD44KO EVs whereas LAMP2b shows no significant changes in both KO cells. We modified the text below reporting the EV phenotypes without specifying subcellular origin of EVs.

After purified from the culture supernatants of CD44KO and CD81KO cells via 100,000 xg ultracentrifugation (Supplementary Figure S6A), the sizes of small EVs (ev44KO and ev81KO) were relatively comparable to the WT control, as characterized by nanoparticle tracking analysis (NTA) and immunoblotting with EV markers (Figure 3E-F, Supplementary Figure S6B). However, when examined by cryo-EM, ev81KO displayed impaired membrane integrity (Figure 3E-F), indicating an essential role for CD81 in modulating EV biogenesis and packaging of membrane proteins.In addition, the EVs are only quantified by the vesicle flow cytometry established by the authors, but then, in functional assay, and in Western blots, EVs are quantified in terms of proteins. It would be important to show if CD44 and CD81 ko also decrease the amount of proteins recovered in the EV preparations, and the number of particles quantified by NTA, as they apparently decrease the number of events detected by vesicle flow cytometry, or not, and if not, why did the authors chose to show only the vesicle flow cytometry results (this is not a very commonly used technic in the field).

Thanks for the suggestion. We have quantified the EVs by vesicle flow cytometry (MFV) and NTA which show consistent EV counts of WT, CD44KO, and CD81 KO (see Author response table 1). We utilize EV protein to normalize the EV education as all the samples had about 6x10^8^ EVs/ µg protein.

**Author response table 1. sa2table1:** 

EV-producing cell	EV counts/mL
NTA	MFV
WT	1.24e+09 +/- 9.44e+07	0.96e+09 +/- 2e+08
CD44KO	1.57e+09 +/- 2.80e+07	1.56e+09 +/- 2e+08
CD81KO	1.62e+09 +/- 1.68e+07	1.75e+09 +/- 9e+07

We have also calculated the EV production data (counts/cell) as measured by NTA in Figure 3- figure supplement 1D which is relatively consistent with the MFV analysis in Figure 3C, demonstrating that CD44KO and CD81KO cells release a higher number of EVs than WT cells.

Another important point to change is the presentation of results as bar graphs: all such graphs must be replaced by graphs showing the position of individual replicates, to illustrate the reproducibility of the presented results (eg Figure 1B, S1C, 3C, 3G-H, 5C, 5E, 5G, 6B, 6H, 6K, etc). (as explained in Weissgerber et al., Plos Biol 2015 13(4): e1002128).

Thanks for the comment and we have updated all bar graphs with raw data points in the updated Figures (Figure 1B, 1C, 3B, 3C, 3F, 3G, 3K, 4E, 4G, 4H, 5C, 5E, 5G, 6B, 6H, 6K, 6N, 6O) and updated figure supplements.

Finally, a somehow frustrating aspect of the paper is that the link between the observed effect of CD44 or CD81 ko on EV release and their in vitro functions on mammosphere formation (Figure 3), and the effect on in vivo tumor growth and metastasis (Figure 5-6) end up as two separate observations (the authors rightly do not claim that impaired exosome release in vivo is responsible for the impaired metastasis). The observation also of clustered CD81 and CD44+ circulating tumor cells in patients (Figure 4) is also somehow a separate observation. Thus there are several stories put together in this article.

We appreciate the comment and apologize for disconnected data presentation. We have reorganized the paper to highlight machine learning-assisted discoveries of CD81 functions and molecular network in partnership with CD44 in promoting cancer stemness which is connected to endocytosis-related EV phenotypes. We admit that in addition to massive manpower and financial support, there are technical limitations in the EV-mediated functional studies in animal models. As proof-of-concept, we therefore utilized the simulated models in vitro to test the hypothesis of EV-CD44 and EV-CD81 in promoting cancer stemness of recipient cells. Our follow-up studies on EV-educated animals are ongoing but beyond the scope of the current manuscript. We are also open to the suggestion leaving the EV part out if that’s recommended by all editors and all reviewers.

Please see updated title “Machine learning-assisted elucidation of CD81-CD44 interactions in promoting cancer stemness and extracellular vesicle integrity” and the abstract.

“Tumor-initiating cells with reprogramming plasticity or stem-progenitor cell properties (stemness) are thought to be essential for cancer development and metastatic regeneration in many cancers; however, elucidation of the underlying molecular network and pathways remains demanding. Combining machine learning and experimental investigation, here we report CD81, a tetraspanin transmembrane protein known to be enriched in extracellular vesicles (EVs), as a newly identified driver of breast cancer stemness and metastasis. Using protein structure modeling and interface prediction-guided mutagenesis, we demonstrate that membrane CD81 interacts with CD44 through their extracellular regions in promoting tumor cell cluster formation and lung metastasis of triple negative breast cancer (TNBC). In-depth global and phosphoproteomic analyses of tumor cells deficient with CD81 or CD44 unveils endocytosis-related pathway alterations, leading to further identification of a quality-keeping role of CD44 and CD81 in EV secretion as well as in EV-associated stemness-promoting function. CD81 is co-expressed along with CD44 in human circulating tumor cells (CTCs) and enriched in clustered CTCs that promote cancer stemness and metastasis, supporting the clinical significance of CD81 in association with patient outcomes. Our study highlights machine learning as a powerful tool in facilitating the molecular understanding of new molecular targets in regulating stemness and metastasis of TNBC.”

Reviewer #2 (Significance (Required)):These results are interesting as showing a novel molecule whose high expression may promote tumor progressions (CD81, as a cluster with CD44). The novelty lies in the functional interaction between CD44 and CD81 leading to the pro-metastatic effect described. Interaction between CD44 and CD81 had been previously observed in a generic proteomic study of EVs (PerezHernandez, D., et al. (2013). J Biol Chem 288: 11649), but not more explored in terms of consequent functions. A pro-metastatic effect of CD81 expression in tumors, especially TNBC, has also been recently demonstrated (Vences-Catalan, F., et al. (2021). Proc Natl Acad Sci U S A 118.). These two articles should be quoted in the current paper.My field of expertise is extracellular vesicles, and their roles in cancer progression. I can only judge superficially the modeling part of the article,

Thank you for highlighting the novelty of CD81 interaction with CD44 as a cluster in promoting tumor progression. We are grateful to the reviewer for providing the CD81 literature information which has been included and cited in the discussion (ref 65, 68), serving as a cross-validation for part of our work.

“A study by Perez-Hernandez et al. also observed CD44 among the EV protein interactome network pulled down by CD81 peptides without exploring their relevance to EV functions (65)…. A potential anti-CD81 therapeutic strategy was identified that may block the pro-metastatic effect of CD81 in animal studies (68).”

Reviewer #3:Major comments:The authors suggest that CD44 and CD81 interact and colocalize inside breast cancer cells. However, staining data presented shows very modest co-localization and immunoprecipitation experiments only employed beads as a negative control. To reinforce their conclusions, the authors should quantify co-localization using standard methods (Mander's or Pearson's coefficients). In addition, the authors should perform negative control immunoprecipitation experiments with antibodies against a target protein not expected to interact with CD81 to show that CD44 binding is specific.

We are grateful for the instructive suggestions. To reinforce our conclusion about the membrane CD44-CD81 interactions, we repeated the Co-IP using anti-CD44 along with two negative controls (new Figure 1F), one of which is the IgG-bead control and the other is CD44KO cell lysate negative control. CD81 was only detected in the protein complex of the WT lysate (TN1 PDX or MDA-MB-231) pulled down by anti-CD44 (new Figure 1F). We also quantified the colocalization using Pearson’s coefficients with average r=0.57 from three different experiments which is now included in Figure 1- figure supplement 2D.

From TEM and quantification in figures 3A, B the authors conclude that there is increased vacuolization with cells. They suggest that purple arrows point to multivesicular endosomes and yellow arrows vacuoles. In fact, the electron dense organelles indicated by purple arrows look more like lysosomes, whereas the yellow arrows appear more like early endosomes/endocytic vesicles. The authors should reassess their vesicle classification system and also provide a breakdown of the proportions of these structures within the graph.

Thanks for the comments. We agree with the reviewer that yellow arrows in Figure 3A could be vacuoles of early endosomes. And we added lysosome images and quantification in Figure 3- figure supplement 1C, showing significant differences among three types of cells (WT, CD44KO, and CD81KO). That may help explain the phenotypes of altered EV release in CD44KO and CD81KO cells.

Given that CD81+/CD44+ EVs are proposed to drive the aggregation and self-renewal of tumor initiating cells, it is somewhat counterintuitive that CD44 and CD81 KO cells show increased levels of EV secretion relative to WT controls. Additionally, it is perplexing that CD44 and CD81 secretion in EVs is unaffected (or my even increase) in knockout cells despite the fact these membrane proteins are supposed to interact and are mutually required for proper expression/localization. How do the authors reconcile these potentially contradictory observations?

We appreciate the diligence and apologize for the lack of quantification and normalized loading in the original western blotting. We have repeated the EV western blots for protein density quantifications three times in new Figure 3E-F that demonstrate a dramatic loss of CD44 in CD81 KO-EVs and a partial loss of CD81 in CD44KO EVs.

The EV rescue experiments in figure 3H, I show recovery of mammosphere formation in CD81 KO cells treated with EVs from WT cells, suggesting that WT EVs are sufficient to rescue selfrenewal. However, it's unclear from their studies whether exosome secretion of CD81/CD44 is actually necessary for aggregation and self-renewal phenotypes. Since CD81 has also been shown to be important for the trafficking of membrane proteins, the loss of self self renewal could relate to cell autonomous alterations in vesicular trafficking in CD81 KO cells. To reconcile between these possibilities, the authors should evaluate how depletion of factors necessary for CD81 secretion (e.g. Rab27a (PMID: 26305877; supplemental data), ESCRT components (PMID: 32049272), or others?) affects mammosphere formation. In either case the results would be extremely interesting and help to determine whether self-renewal is controlled by CD81 via cell autonomous or non-cell autonomous mechanisms.

We are thankful for the extremely intriguing question about cell autonomous and non-cell autonomous roles of CD81 in controlling self-renewal. To address the question, we did transfect siRNAs to knock down Rab27a levels in TNBC cells and found a decreased efficiency in mammosphere formation of these cells in comparison to scrambled (Scr) control cells (see Author response image 3), suggesting a possible non-cell autonomous mechanism of CD81 promoted self-renewal.

**Author response image 3. sa2fig3:** CD81 is required for EV-induced effects on stemness signature proteins and mammosphere formation. Representative images (left two panels) and quantifications (right panel) of mammospheres (Day 5) derived from 500 MDA-MB-231 cells in suspension after transfections with scrambled (Scr) RNAs and human siRab27a.

The authors observe that CD81 depletion profoundly impairs primary tumor development and metastasis in pre-clinical models. In fact, the defect in metastasis appears to be secondary to the robust impairment in tumorigenesis. What is the fate of CD81 KO cells in mammosphere assays and transplant models? Do CD81 KO cells have reduced viability and/or proliferation in vitro and in vivo? Can the defects in mammosphere formation and tumorigenesis in CD81 KO cells be rescued via re-introduction of wtCD81? What about mutant CD81 that is deficient for CD44 binding? These studies will help to delineate the role of CD81 in primary tumor development and whether interaction with CD44 is require for this process.

Thanks for the comments and questions. We included the data on slightly slower cell proliferation of human CD81KO and mouse Cd81KO cells (no obvious cell death) in Figure 1- figure supplement 1E-F. We have also reintroduced wtCD81 in two sets of distinct vectors into CD81KO cells and observed wtCD81 in both HA and GFP vectors rescued the defects of CD81KO cells while mutant CD81 deficient for CD44 binding failed to do so, demonstrating the role of CD81CD44 interaction in promoting self-renewal.

In the authors model, they show CD81 interactions with CD44 facilitating EV secretion which enhances self-renewal and CD44 alone facilitates tumor cell clustering. However, they show that CD81 can also binding CD44. Does the CD81-CD44 interaction also serve to facilitate clustering between tumor cells?

Yes, we conducted the tumor cell clustering experiment and added the new Figure 4- figure supplement 4D which demonstrates that CD81 WT rescues the clustering of CD81KO cells and the CD81 truncated mutant does not.

Minor Points:In a number of figures (e.g. Fig, 1C; Figure 3E, H) the authors show representative immunoblots but there are no indications in the legends of how many times the experiment was performed. Presumably at least 3 independent experiments were performed. The authors should also include quantification of these data to support that these effects are reproducible and significant. All data points should be plotted within graphs so that the reader can note the distribution of the data.

Thanks for the suggestions. We have included all raw data points in all quantified bar graphs and quantified the western blots from at least three independent experiments.

While the manuscript was generally well written, there are a couple of grammatical mistakes that can be fixed.

We appreciate the positive comment and have corrected the grammatical mistakes by native /professional writers.